bioengineering/biomechanics

exotendon, kinematics, kinetics, muscle activation

**Authors for correspondence:**
Caihua Xiong
e-mail: chxiong@hust.edu.cn
Jiejunyi Liang
e-mail: jjy_liang@hust.edu.cn

# A portable exotendon assisting hip and knee joints reduces muscular burden during walking

Longfei Cheng, Caihua Xiong, Wenbin Chen, Jiejunyi Liang, Bo Huang and Xiaowei Xu

Institute of Rehabilitation and Medical Robotics, State Key Lab of Digital Manufacturing Equipment and Technology, Huazhong University of Science and Technology, Wuhan, Hubei, People's Republic of China

LC, 0000-0002-2533-3552; CX, 0000-0003-2326-0289;
WC, 0000-0002-2273-1934; JL, 0000-0001-8549-0261;
BH, 0000-0001-7168-4495

Assistive devices are used to reduce human effort during locomotion with increasing success. More assistance strategies are worth exploring, so we aimed to design a lightweight biarticular device with well-chosen parameters to reduce muscle effort. Based on the experience of previous success, we designed an exotendon to assist in swing leg deceleration. Then we conducted experiments to test the performance of the exotendon with different spring stiffness during walking. With the assistance of the exotendon, peak activation of semitendinosus decreased, with the largest reduction of 12.3% achieved with the highest spring stiffness ($p = 0.004$). The peak activations of other measured muscles were not significantly different ($p = 0.15$–$0.92$). The biological hip extension and knee flexion moments likewise significantly decreased with the spring stiffness ($p < 0.01$). The joint angle was altered during the assisted phases with decreased hip flexion and knee extension. Meanwhile, the step frequency and the step length were also altered, while the step width remained unaffected. Gait variability changed only in the frontal plane, exhibiting lower step width variability. We conclude that passive devices assisting hip extension and knee flexion can significantly reduce the burden on the hamstring muscles, while the kinematics is easily altered.

## 1. Introduction

Humans have evolved with muscular and skeletal systems suitable for locomotion, such as long Achilles tendons [1], short

toes [2] and elastic foot arches [3], which makes it challenging to develop assistive devices. Although the earliest assistive device dates back to the 1890s [4], success in reducing the metabolic cost of locomotion has not been achieved until recently. The success in the recent development of assistive devices is not an accident, and it is based on the knowledge of human anatomy, biomechanics and physiology of locomotion. Generally, effective assistance requires coordinated interaction between the human and the device. The device provides appropriate assistive torque at joints, while the human body makes adjustments, such as muscle recruitment, hence making use of the assistance. The comprehensive result of an assistive device includes the benefits from the external assistance and the penalty from the added mass and possible kinematic constraints. Therefore, improving the device design from these two perspectives is the key to augmenting human locomotion ability.

Researchers try to maximize the benefits from external assistance mainly in two ways, i.e. applying assistance to subtasks of effort, such as redirection of the centre of mass, and selecting appropriate assistance parameters. Specifically, the focus is mainly on the ankle and hip joints, as these two joints contribute most to the positive work during walking and running. In 2013, Malcolm *et al.* [5] developed an active ankle exoskeleton which reduced the metabolic cost of walking by 6% compared with using no device. In 2015, Collins *et al.* [6] designed a passive exoskeleton to assist in ankle plantarflexion. The energy cost decreased by 7.2% compared with that of normal walking. Subsequently, numerous researchers also obtained a lower metabolic cost of walking or running using different structures to assist in ankle plantarflexion [7–9]. The efficiency of hip muscles is thought to be lower than that of ankle muscles, attracting increasing attention on hip assistance [10]. The metabolic cost of walking or running decreased significantly, irrespective of whether the device assists hip flexion [11,12], hip extension [13,14] or both [15]. Additionally, appropriate assistance parameters also play an important role in maximizing the benefits from the external assistance. Previous studies attempted to find the optimal performance of devices using different timings [8,16,17] and magnitudes of assistance [6,18]. Humans can be also included in the optimization process, and device control is varied to minimize the real-time measured energy cost, i.e. human-in-the-loop optimization. This approach can customize assistance parameters for individuals and has been applied to the ankle and hip exoskeletons, both of which have significantly reduced the metabolic cost of walking [19,20].

Furthermore, knowledge regarding human anatomy characteristics aids in the design of assistive devices. Biarticular muscles are common in humans and animals. Similar to the monoarticular muscles, biarticular muscles support energy storage and return. They also promote the energy transfer from the proximal to the distal joint. In addition, the feature that biarticular muscles flex and extend the adjacent joints leaves the muscle fibres working near the optimal length. Therefore, they play an important role in improving efficiency during human locomotion [21,22]. In view of these facts, biarticular or multiarticular designs are highly anticipated. Van den Bogert [23] proposed four designs of exotendon and demonstrated, in a simulation, that appropriate arrangements of elastic elements could reduce muscle forces required for walking. A biarticular knee–ankle exoskeleton achieves a higher reduction in the metabolic cost than the monoarticular assistance [24]. Biarticular passive devices usually arrange elastic elements parallel to the distribution of muscles, mimicking the function of human biarticular muscles, such as gastrocnemius [25] and hamstring muscles [26,27]. By contrast, non-bionic multiarticular designs are usually applied in active devices during normal walking [18], loaded walking [28] and sitting transfers [29]. These bioinspired designs have successfully reduced human effort.

The penalty from the assistive device mainly originates from the added mass and the possible kinematic constraints. The rigid frame is one of the reasons for the added mass and kinematic constraints. The kinematic constraints are mainly caused by the inconsistent joint axes between the human and the device. It may also cause stress concentration and thus cause discomfort or even physical damage to users [30]. Therefore, in recent years, soft exosuits have been developed with no rigid structures and using cables to transfer assistive forces, showing great potential for augmenting human locomotion ability [14,28,31,32]. External power sources are the other reason for the added mass for active devices. By contrast, passive devices lack external energy sources, and thus they are more lightweight. Passive devices have been developed to assist the ankle and hip joints, and the walking or running economy has been improved by 6.4–8.0% [6,12,15,33,34]. Additionally, distributing the added mass proximally also contributes to improving the walking economy [35], which may be one of the reasons for recent popular hip movement assistance.

To explore more assistance strategies, assistance in subtasks of not too much effort, such as swing leg deceleration, is also popular. Simpson *et al.* [33] connected the legs with a spring to aid in the deceleration, and they achieved a more metabolic reduction than expected due to the complex

interaction between assistance dynamics and human adaption strategies. Barazesh & Sharbafi [26] found lower muscle activation and metabolic cost using length-modulated compliant hip-based gait assistance via a neuromuscular model, and they also experimentally verified that the support to the biarticular muscles on the thigh reduced the metabolic cost by 4.68%. Removing the energy from the swing leg deceleration and applying the energy to generating electricity improved the walking economy by 2.5% [36]. If the energy stored from the deceleration was used to assist the hip extension during the early stance phase with the help of a clutch, the walking economy was improved further up to 8.6% [27].

Although researchers have achieved significant successes, there remain some challenging problems. Unlike flexible controllability of active assistive devices, assistance parameters of passive devices depend extensively on the physical parameters as well as human movement. Appropriate parameters are important, especially for biarticular designs. Biarticular devices actuate two joints meanwhile, whereas the required assistance for the two joints may differ. Therefore, selecting the appropriate physical parameters to achieve theoretical assistance is difficult. Further, due to the complexity of the biological system, the assistance is not always effective as expected. A passive multiarticular exoskeleton was designed to assist hip flexion and ankle plantarflexion [37]. The biological joint moment decreased significantly, but the metabolic cost increased. The added mass was not sufficient to explain the increase, as the metabolic cost in the assistance condition was higher than that in the condition of wearing the exoskeleton with no assistance. Therefore, understanding the interaction between the human and the device further remains necessary.

In this study, we attempted to maximize the performances from the aforementioned aspects. We aimed at developing a lightweight biarticular device with appropriate parameters to assist the swing leg deceleration and investigating the consequent effects of our device on muscle activation, kinematics and joint kinetics. The exotendon was arranged posterior to the thigh, acting as a partial role of hamstring muscles to assist the swing leg deceleration. To reduce the added mass and ensure user's comfort, we did not use any rigid frame. By contrast to the study of Barazesh & Sharbafi [26], we only assisted the hamstring muscles and put more focus on the effect of the assistance on human biomechanics experimentally. We first conducted a preliminary experiment to make clear the assistance characteristics and then obtained the physical parameters via optimization based on the walking kinematics and kinetics, as in previous studies [23,38]. Finally, we examined the effects of different assistance levels on muscle activation, kinematics and kinetics. We expected lower muscle activation when replacing the partial function of the hamstring muscles with our device. We also expected that the kinematics remained unchanged by adopting a lightweight design and appropriate physical parameters.

# 2. Methods

## 2.1. Bioinspired design

Humans actively swing their legs forward during most of the swing phase while walking, but in a short time before the heel-strike, they will slow down the leg and even move it in the opposite direction. This phenomenon is known as swing leg retraction, which provides some potential benefits, such as improved walking stability and reduced heel-strike impact [39]. However, to achieve these benefits, hamstring muscles, in charge of the hip extension and knee flexion, have to be activated to produce the required joint moments to brake forward swing of the leg [40]. Hence, the retraction is of effort.

Inspired by the phenomenon and earlier passive devices, we designed an exotendon that plays a partial role of the hamstring muscles. The exotendon is composed of springs, inelastic webbing, Velcro, ankle straps and a waist belt (figure 1). The material of the webbing is polyester fibres. Ankle straps and the waist belt were adapted from commercial ankle supports and waist support belts, respectively. Ankle supports are made from styrene–butadiene rubber and nylon. The waist support belt is made from styrene–butadiene rubber and Lycra. The spring is connected to the webbing by Velcro, and the other end of the webbing is sewn to the waist belt or the ankle strap. To minimize movement of the device resulting from assistive forces and increase the moment arms relative to the hip and knee joints, the waist belt is located above the ilium, and the ankle strap is located above the lateral malleolus. Physical parameters of the device include the stiffness and the resting length. The stiffness is adjusted by selecting springs of different stiffness, and the resting length is adjusted by the Velcro. The total mass is 370–400 g, and the distal mass is only 103 g.

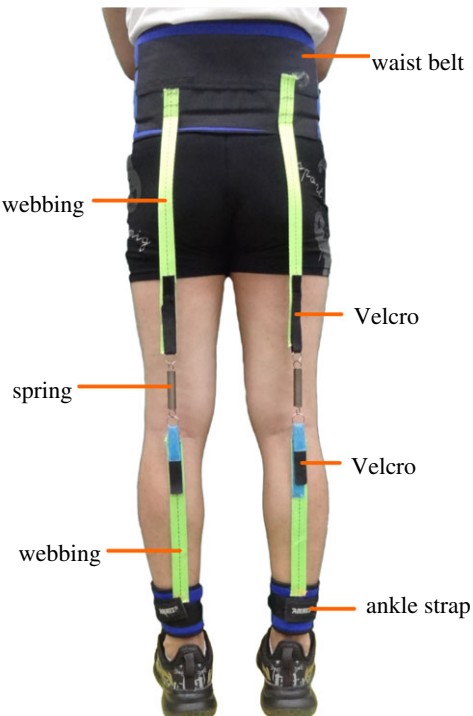

**Figure 1.** Posterior view of the exotendon worn by a subject. The exotendon was composed of springs, inelastic webbing, Velcro, ankle straps and a waist belt.

## 2.2. Determination of physical parameters of the exotendon

Assistance parameters have a significant influence on the performance of the device [8,16,17]. Previous studies on passive devices have shown the effects of physical parameters on the performance, such as the resting length [25] and spring stiffness [6]. In this study, the exotendon lengthens during the late swing phase and shortens gradually after the heel-strike. Therefore, the exotendon can provide the hip extension and knee flexion torques during the late swing phase and the early stance phase. During the phases, the joint moment is the extension moment at the hip and the flexion moment at the knee [40], and thus the assistive torque can partially replace the joint moment produced by muscles. However, the time range of the hip extension moment is not entirely consistent with that of the knee flexion moment [40], such that the exotendon will provide resistance to the knee movement when assisting the hip during certain phases. Therefore, the physical parameters of the exotendon need to be determined carefully.

The biological joint moment is the net moment produced by the muscle force, and thus reducing it helps reduce muscle effort. Based on the principle of minimizing the total biological joint moment, optimization was performed to determine the physical parameters of the exotendon. To perform the optimization, we assumed that the exotendon did not affect the joint kinematics and kinetics of lower limbs. The other assumption was that muscle effort increased with the biological joint moment.

The assistive force $F(t)$ produced by the exotendon at a certain moment $t$ during walking is calculated by Hooke's law, i.e.

$$F(t) = \begin{cases} k(l(t) - l_0) & l(t) > l_0 \\ 0 & l(t) \leq l_0 \end{cases}, \tag{2.1}$$

where $k$ is the spring stiffness, $l_0$ is the exotendon resting length, and $l(t)$ is the exotendon length at the moment $t$. The assistive torque produced by the force $F(t)$ at the hip and knee, i.e. $\tau_{\text{exo\_hip}}(t)$ and $\tau_{\text{exo\_knee}}(t)$, can be calculated as

$$\tau_{\text{exo\_hip}}(t) = F(t) \cdot R_{\text{hip}}(t) \tag{2.2}$$

and

$$\tau_{\text{exo\_knee}}(t) = F(t) \cdot R_{\text{knee}}(t), \tag{2.3}$$

where $R_{\text{hip}}(t)$ and $R_{\text{knee}}(t)$ are the moment arms relative to the hip and knee joint centres, respectively. Then, the biological joint moments are obtained by subtracting the assistive torque from the joint moments, similar to a previous study [23]. So we can get the biological joint moments at the hip and knee joints, i.e. $\tau_{\text{bio\_hip}}(t)$ and $\tau_{\text{bio\_knee}}(t)$,

$$\tau_{\text{bio\_hip}}(t) = \tau_{\text{hip}}(t) - \tau_{\text{exo\_hip}}(t) \tag{2.4}$$

and

$$\tau_{\text{bio\_knee}}(t) = \tau_{\text{knee}}(t) - \tau_{\text{exo\_knee}}(t), \tag{2.5}$$

where $\tau_{\text{hip}}(t)$ and $\tau_{\text{knee}}(t)$ are the joint moments of the hip and knee, respectively, calculated by inverse dynamics. Because the exotendon mainly acts on the hip and knee joints, the ankle joint is considered to be almost unaffected. Then, the objective function $C$ is calculated as the total biological joint moment of the hip and knee in a gait cycle time $T$, i.e.

$$C = \int_0^T |\tau_{\text{bio\_hip}}(t)| \, dt + \int_0^T |\tau_{\text{bio\_knee}}(t)| \, dt. \tag{2.6}$$

Joint moments, exotendon length change in a gait cycle and moment arms can be measured experimentally, leaving the exotendon resting length and the spring stiffness as the undetermined parameters. According to the above calculation, the resting length affects the timings and magnitudes of the assistive forces, and short resting length will cause large time ranges and magnitudes. The spring stiffness only affects the assistance magnitudes. If the spring stiffness is not limited, the optimized spring stiffness may produce comparable assistive torque to the human joint moment during assisted phases. Given that too large assistive forces may offset the benefits, we did not optimize the spring stiffness and only selected three kinds of spring stiffness, i.e. 0.3, 2.2 and 4.1 kN m$^{-1}$. We performed optimization via function *ga* in Matlab software (MathWorks, Natick, MA, USA) to determine the exotendon resting length for each spring stiffness.

## 2.3. Experimental testing

Eleven healthy male adults (age 24.2 ± 1.7 years, weight 67.8 ± 6.2 kg, height 1.73 ± 0.05 m) volunteered to participate in the experiment. All subjects had no musculoskeletal injuries and were asked not to perform any strenuous exercise for 24 h prior to testing.

The experiment comprised two sessions performed on two different days. The first session was performed to obtain the data required to determine the resting length for each spring stiffness. Subjects were instructed to walk on an instrumented treadmill (Tandem, AMTI, Watertown, MA, USA) with the exotendon at 1.4 m s$^{-1}$ for 5 min. In order to minimize the effect of assistive forces on human movement, the spring stiffness selected was only about 200 N m$^{-1}$. To ensure that the measurement of the moment arm was correct, the exotendon must be kept in tension in the target time range. We let subjects maintain a pose similar to that in the middle stance phase and kept the exotendon slightly preloaded.

The second session was the formal testing, including four walking trials, i.e. a control condition (CON) and three conditions wearing the exotendon with a spring stiffness of 0.3, 2.2 and 4.1 kN m$^{-1}$, respectively (EXO03, EXO22, EXO41). Given that two retroreflective markers were attached to the waist belt and the device was lightweight, we selected wearing the device without springs as the CON to minimize the effects of moving markers on the kinematics and kinetics. For each spring stiffness, the resting length was obtained by the aforementioned optimization. Subjects walked on the treadmill at 1.4 m s$^{-1}$ for 5 min. The order of four conditions was randomized to minimize the effects of learning and fatigue. Before testing, subjects were asked to walk on the treadmill for 20 min to familiarize themselves with the device and experimental conditions.

## 2.4. Data collection and analyses

In the first session, we collected data, including joint moments, exotendon length change in a gait cycle, and moment arms relative to the hip and knee joints. A motion capture system with nine cameras (Vicon, Oxford, UK) was used combined with an instrumented treadmill embedded with two force plates. The sampling rates of the cameras and force plates were 100 and 1000 Hz, respectively. Sixteen retroreflective markers were attached to the body according to the plug-in gait lower body model. The raw trajectories

were filtered using the Woltring fifth spline algorithm with a mean squared error of 10. The joint moment was calculated by inverse dynamics via the Nexus software (Vicon, Oxford, UK). The positive moment is the extension moment for the hip and knee joints or the plantarflexion moment for the ankle joint. Two retroreflective markers were attached to the two attachment points of the exotendon. The exotendon length was calculated as the distance between these two markers. Meanwhile, two markers were attached to the great trochanter and lateral epicondyle of the femur representing joint centres of the hip and knee, and two other markers were attached to the two ends of the spring to determine the direction of the force. Then, the moment arm was calculated from the joint rotation centre and the line of action of the exotendon force. We took half-minute data during the last minute of trials to process. All data were normalized to a gait cycle. The gait cycle was segmented according to the ground reaction forces with a threshold of 20 N.

In the second session, the assistive torque, electromyography (EMG), kinematics and joint kinetics were measured. In three conditions wearing the exotendon with springs, a single-axis load cell (Forsentek, Shenzhen, Guangdong, China) was in series with the spring on the right side to measure the forces that the exotendon exerted on the body, and the sampling rate was 1000 Hz. The raw signals were filtered with a Butterworth filter with a cut-off frequency of 20 Hz. Measurements of moment arms were the same as those in the first session. Then, the assistive torque can be calculated by multiplying the force and the moment arm. The EMG signal was measured by an EMG system (Biometrics, Newport, UK) with a sampling rate of 1000 Hz. The measured muscles included soleus (SOL), gastrocnemius medialis (GAS), vastus medialis (VM), rectus femoris (RF), semitendinosus (SEM) and gluteus maximus (GM). The electrodes were attached to the right leg, and the position of the electrodes was determined according to SENIAM guidelines [41] or by palpation. We elaborately prepared the subjects' skin, including shaving and wiping with alcohol. The raw signals were amplified by an amplifier with a bandwidth of 20–460 Hz. Then, the data were rectified and filtered with a zero-lag Butterworth filter with a cut-off frequency of 6 Hz to obtain the linear envelope. Next, the envelope was normalized to the maximal value in the CON condition. Finally, the data were divided for each gait cycle and time-normalized from 0% to 100%. Two statistics describing the extent of muscle activation, the root mean square (RMS) value and the peak value of normalized EMG, were calculated. Notably, the peak value of RF is the maximum during the stance phase. Kinematic data included joint angles and gait parameters. Sixteen retroreflective markers were determined according to the plug-in gait lower body model to calculate joint angles using the Nexus software. Joint range of motion (ROM) was calculated as the difference between the maximal flexion or dorsiflexion and the maximal extension or plantarflexion during a gait cycle. The step frequency (SF) was obtained from the reciprocal of time between two successive heel-strikes. The step length (SL) was calculated as the distance between two heel markers along the forward direction when the heel-strike occurred. The step width (SW) was calculated as the maximal distance between two heel markers along the medial–lateral direction during the double support phase [42]. Gait variability was expressed as standard deviations of the gait parameters. For example, the SF variability was calculated as the standard deviation of SF data from the selected time range. Likewise, the joint moment was also calculated by inverse dynamics via Nexus software. The total joint extension or plantarflexion moment was calculated as the numerical integration of the joint extension or plantarflexion moment over the whole gait cycle. The total flexion or dorsiflexion moment was calculated in a similar way.

## 2.5. Statistics

All data analyses were performed in the Matlab software. We focused on joint angles and moments in the sagittal plane only, where the joint movement mainly occurred. One-way repeated measures analysis of variance (ANOVA) was applied to determine the difference in the EMG RMS value and peak value, gait parameters (i.e. SF, SL and SW) as well as their variability, joint ROM, maximal joint flexion or dorsiflexion, maximal joint extension or plantarflexion, the total joint extension or plantarflexion moment and the total joint flexion or dorsiflexion moment among four conditions. Before ANOVA, Mauchly's sphericity test was applied to test for the sphericity assumption. For those not satisfying the assumption, $p$-values resulting from ANOVA were corrected by the Greenhouse–Geisser method. If the result from ANOVA showed significant differences, further multiple comparisons were performed by paired $t$-tests for normally distributed data and two-sided Wilcoxon signed-rank tests for non-normally distributed data. The normality of data was examined using Lilliefors tests. The significance level was set at $\alpha = 0.05$. Data are represented as mean ± s.d. unless otherwise stated.

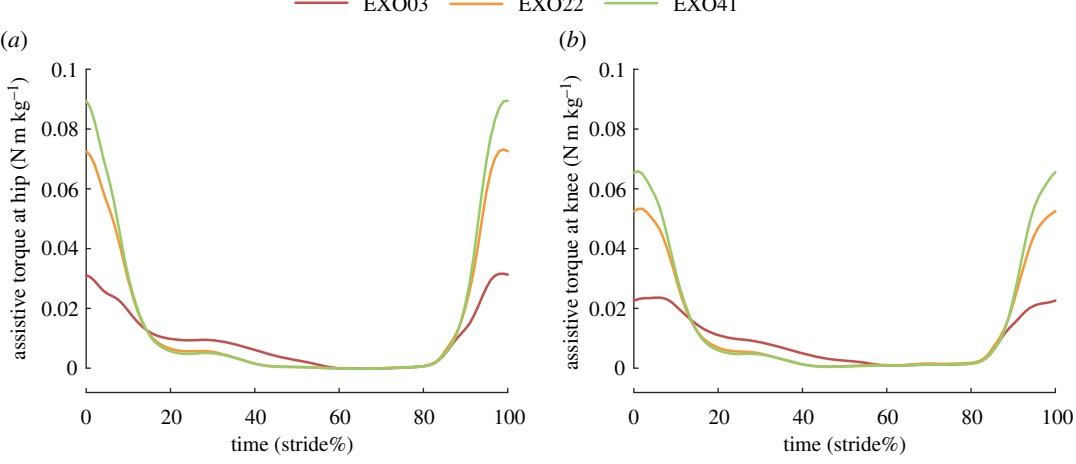

**Figure 2.** Assistive torque at the hip (*a*) and knee (*b*) joints during a gait cycle under three conditions with assistance.

# 3. Results

## 3.1. Assistive torque

The assistive torque at the hip and knee joints under three conditions with assistance is shown in figure 2. The assistance started at approximately 85% of a gait cycle and ended at 18% of the next one for the EXO22 and EXO41 conditions. The time range of the assistance in the EXO03 condition was larger, whereas the main range was similar among three conditions. The peak assistive torque during a gait cycle provided at the hip and knee joints appeared around the heel-strike for all three conditions with assistance. The exotendon provided the highest assistance magnitude in the EXO41 condition, with peak values of 0.090 N m kg$^{-1}$ at the hip joint and 0.066 N m kg$^{-1}$ at the knee joint, which accounted for about 14.6% and 12.8% of the hip and knee peak joint moments during the assistance phases, respectively.

## 3.2. Muscle activation

Some data were excluded as outliers due to the electrode's malfunction. Activation of the measured muscles during a gait cycle is shown in figure 3, and the corresponding RMS and peak values are shown in figure 4. The activation of SEM showed significant differences among four conditions ($N = 10$; ANOVA; $p < 0.01$ for the RMS and peak values; figure 4$e$). In detail, the muscle activation of SEM decreased with the spring stiffness, and in the EXO41 condition, the RMS value decreased by $11.1 \pm 8.2\%$ ($N = 10$; paired $t$-test; $p = 0.002$) and the peak value decreased by $12.3 \pm 10.1\%$ ($N = 10$; paired $t$-test; $p = 0.004$). The RMS and peak values of GM were not significantly different among conditions ($N = 9$; ANOVA; $p = 0.31$, 0.15), but the peak value had a tendency to decrease in EXO22 and EXO41 conditions (figure 4$f$). Also of note, muscle activation of RF showed a visible difference, though the differences in RMS values and peak values were not significant ($N = 10$; ANOVA; $p = 0.11$, 0.19; figure 4$d$). The peak activation decreased slightly during the assisted phases, while the activation increased with the spring stiffness during the unassisted phases (figure 3$d$). Similarly, the activation of VM also showed a slightly decreasing trend during the assisted phases (figure 3$c$), but the statistical results of the RMS and peak values were not significant ($N = 9$; ANOVA; $p = 0.67$, 0.61; figure 4$c$). The RMS and peak values of SOL and GAS were not significantly different among conditions ($N = 11$; ANOVA; $p = 0.36$–0.92; figure 4$a,b$).

## 3.3. Kinetics

Data of one subject were excluded as outliers. Biological joint moments in a gait cycle and the total biological joint moment are shown in figure 5. The hip and knee biological joint moment curves showed visible differences among conditions. The total hip extension moment was reduced significantly with the assistance of the exotendon ($N = 10$; ANOVA; $p < 0.01$; figure 5$a$), and the reduction increased with the spring stiffness. In the EXO41 condition, the total hip extension moment

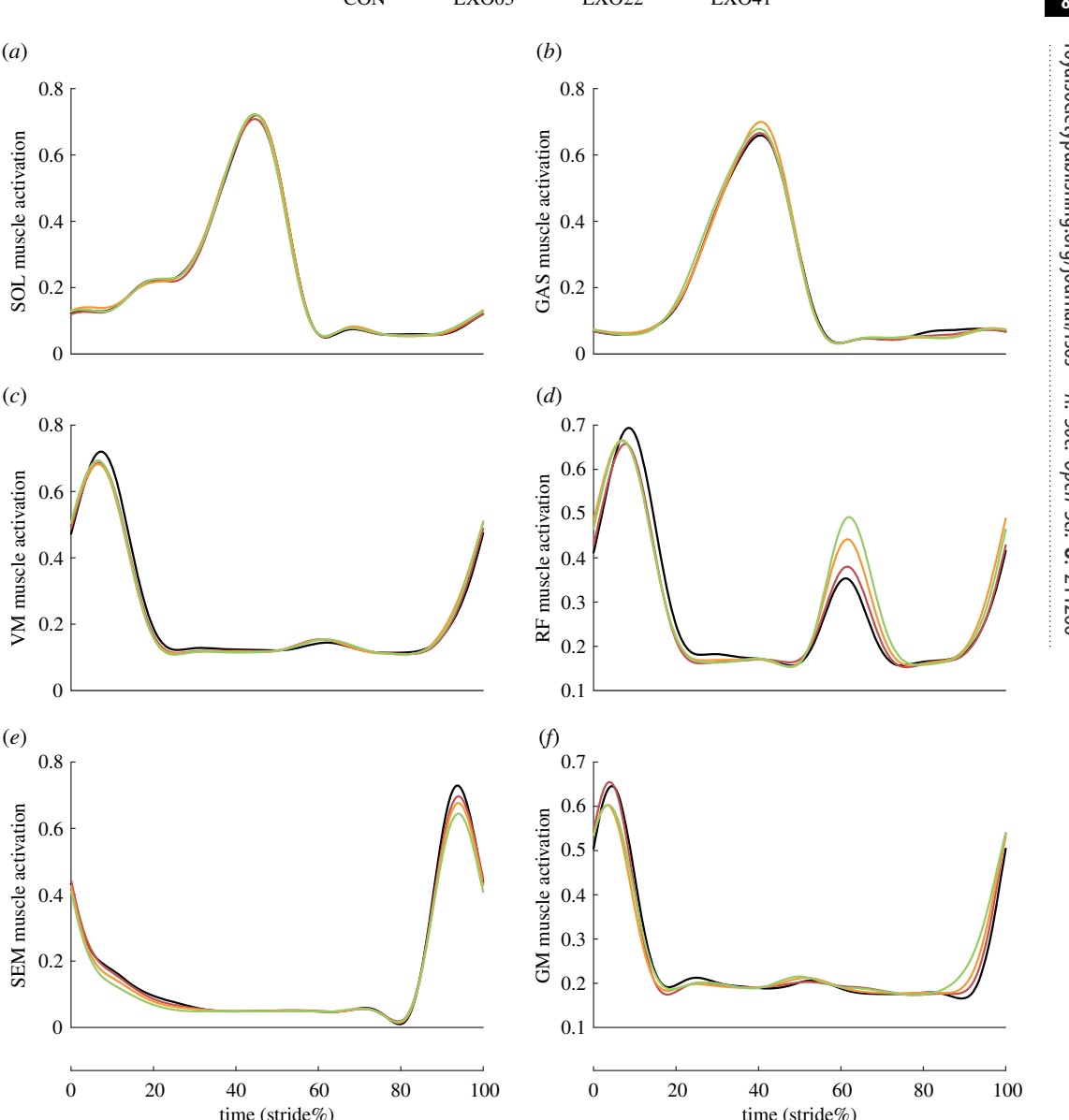

**Figure 3.** Muscle activation during a gait cycle of six lower limb muscles under four conditions.

decreased by about 12.6 ± 4.3% (*N* = 10; paired *t*-test; *p* < 0.01). Although the difference in total hip flexion moment was not significant, there was a tendency that exotendon with high spring stiffness increased the hip flexion moment (*N* = 10; ANOVA; *p* = 0.08; figure 5*a*). By contrast, the total knee flexion moment was significantly reduced with the exotendon (*N* = 10; ANOVA; *p* = 0.007; figure 5*b*), and the highest reduction occurred in EXO41 condition by 10.3 ± 10.2% (*N* = 10; paired *t*-test; *p* = 0.01) while the total knee extension moment was not significantly changed (*N* = 10; ANOVA; *p* = 0.14; figure 5*b*). Notably, the total plantarflexion moment decreased significantly (*N* = 10; ANOVA; *p* < 0.01; figure 5*c*), and the highest reduction occurred in the EXO41 condition by 7.5 ± 3.9% (*N* = 10; paired *t*-test; *p* < 0.01). The total ankle dorsiflexion moment was not significantly different among conditions (*N* = 10; ANOVA; *p* = 0.15; figure 5*c*).

## 3.4. Kinematics

The exotendon had a partial effect on kinematics. As for the joint angle, less hip flexion and knee extension were observed during the assisted phases, and more hip extension and less knee flexion

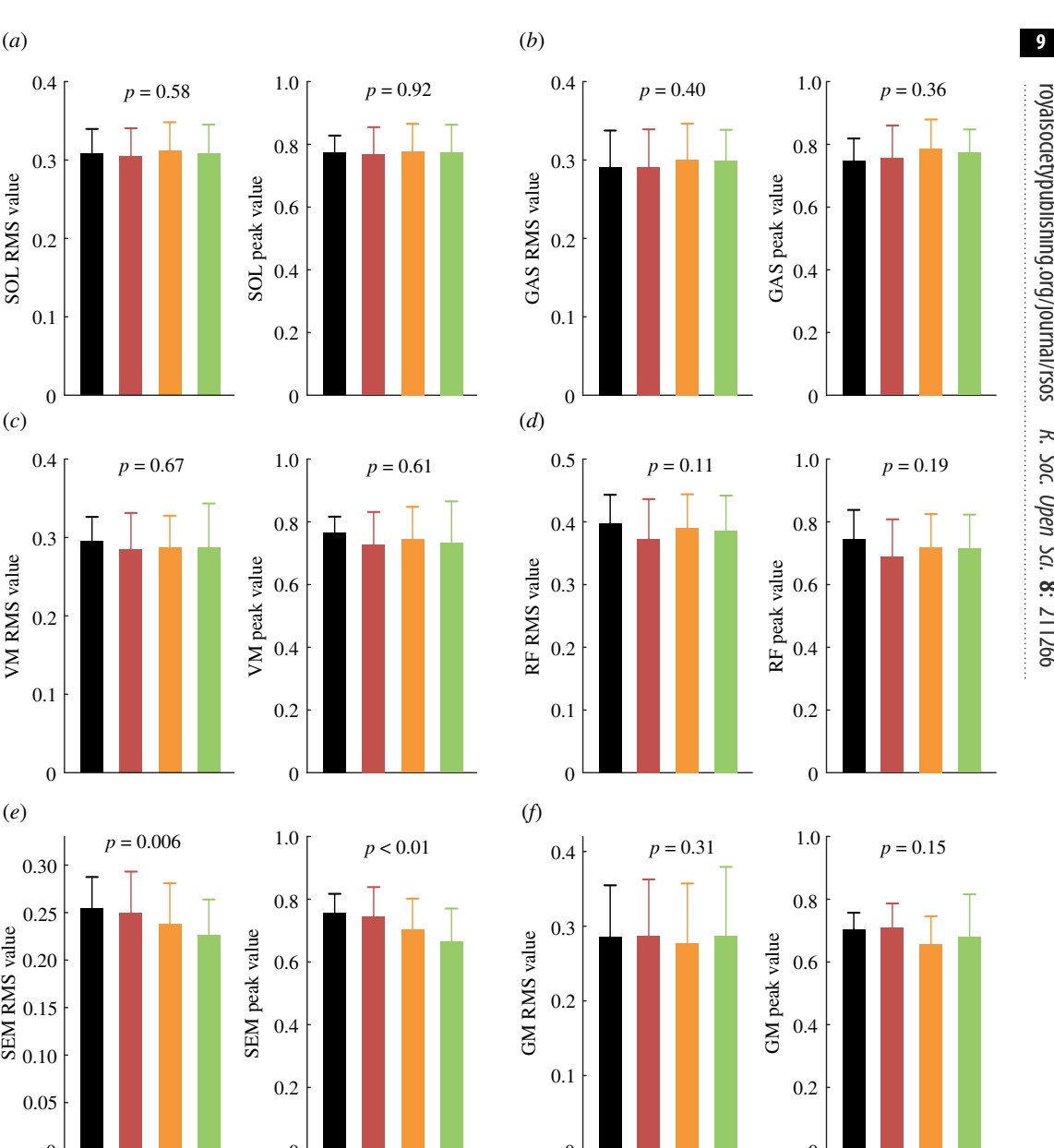

**Figure 4.** RMS and peak values of six lower limb muscles under four conditions. Bar represents the mean value across subjects, and error bar represents mean ± s.d.

were observed during the unassisted phases (figure 6*a*,*b*). Specifically, the maximal flexion of the hip decreased with the spring stiffness ($N = 11$; ANOVA; $p < 0.01$), and in the EXO41 condition it decreased by about 5.1° ($N = 11$; paired *t*-test; $p < 0.01$). The maximal extension of the hip increased with the spring stiffness ($N = 11$; ANOVA; $p < 0.01$; figure 6*a*), and in the EXO41 condition it increased by about 4.8° ($N = 11$; paired *t*-test; $p < 0.01$). However, in total, the hip ROM was not significantly changed (table 1). The maximal flexion and the maximal extension of the knee decreased slightly with the spring stiffness ($N = 11$; ANOVA; $p = 0.02$, 0.03, respectively; figure 6*b*), and the reduction was about 1.9° in the EXO41 condition ($N = 11$; paired *t*-test; $p = 0.004$, 0.04, respectively). In total, the knee ROM significantly decreased by about 3.8° (table 1). The maximal plantarflexion, the maximal dorsiflexion and the ROM of the ankle did not show significant differences among conditions ($N = 11$; ANOVA; $p = 0.34$–0.87; figure 6*c*).

SF and SL were significantly different among conditions, and the exotendon with higher spring stiffness caused larger SF and shorter SL (table 1). However, SW was similar among four conditions

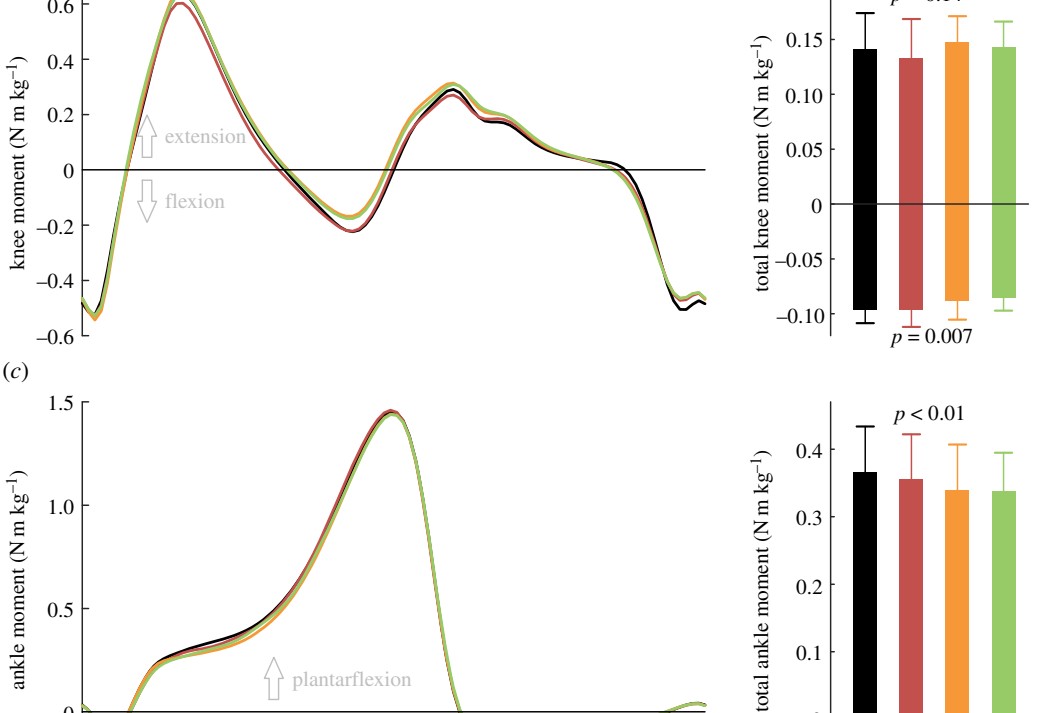

**Figure 5.** Biological joint moment and the total biological joint moments during a gait cycle of three joints under four conditions. Bar represents the mean value across subjects, and error bar represents mean ± s.d.

(table 1). By contrast, the SF variability and the SL variability were not significantly different while the SW variability was significantly reduced (table 1).

## 4. Discussion

In this study, we designed a lightweight biarticular exotendon to assist hip extension and knee flexion and examined its performance under different physical parameters. We found that compared with the CON condition, wearing the exotendon significantly reduced activation of the semitendinosus, and the highest reduction occurred in the highest spring stiffness. As expected, the biological joint moments of the hip and knee decreased during the assisted phases. However, the kinematics was also significantly altered, where the hip flexion and the knee extension decreased during the assisted phases.

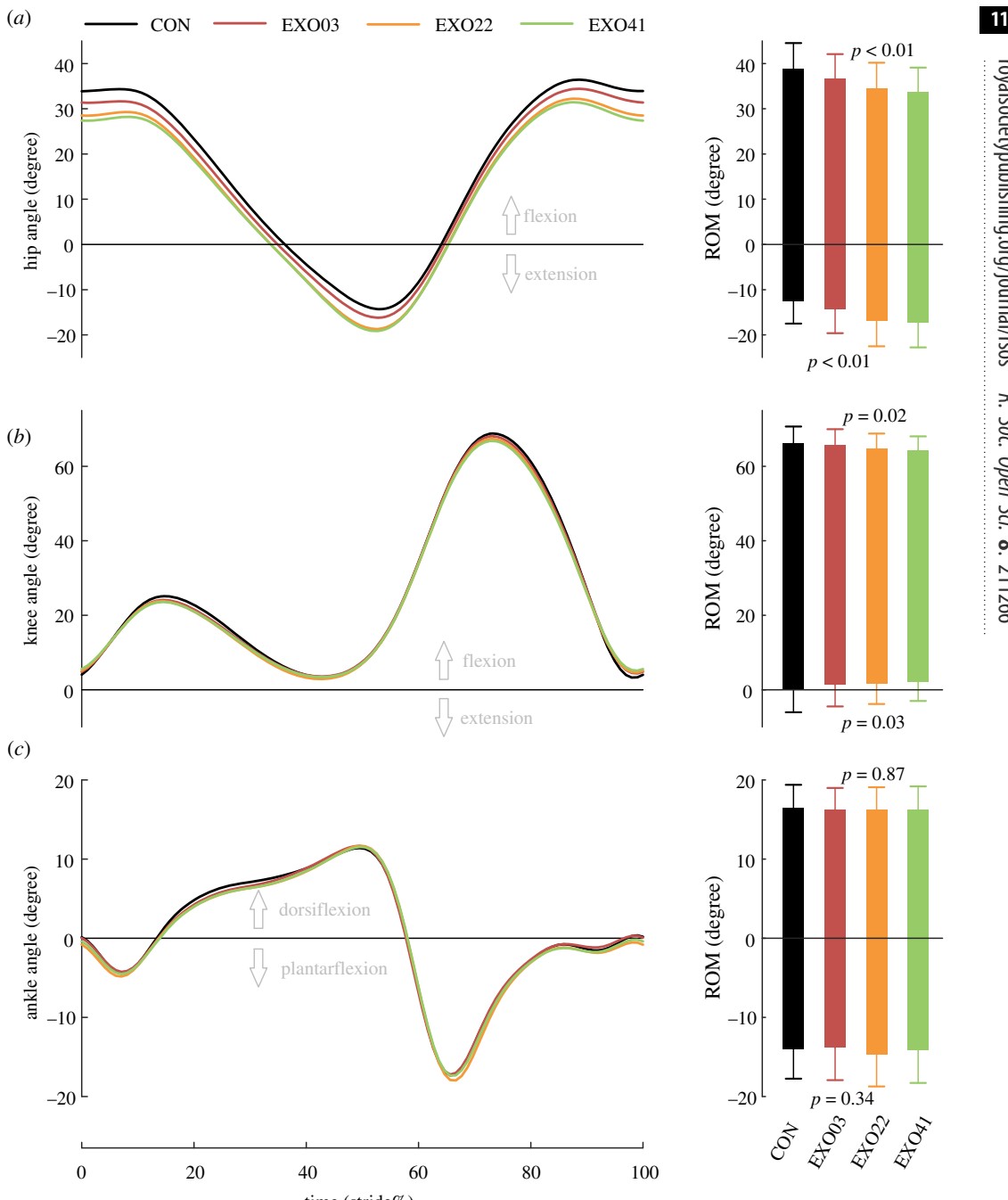

**Figure 6.** Joint angles and the corresponding range of motion during a gait cycle of three joints under four conditions. Bar represents the mean value across subjects, and error bar represents mean ± s.d.

Humans retract their legs during the late swing phases to prepare for the upcoming heel-strike. To this end, the hamstring muscles posterior to the thigh must be activated to produce the required forces. The exotendon presented in this study imitates the function of the hamstring muscles and can replace part of the forces produced by muscles. The results showed that the exotendon worked well in acting as a partial role of the hamstring muscles. The RMS and peak values of SEM decreased significantly in all three conditions with assistance. Because the human body is not a purely mechanical system, the interaction between the human body and the assistive devices is complicated. In this study, although the RMS values of VM and RF, anterior to the thigh, were not significantly different, they exhibited a trend of lower activation during the assisted phases, while activation of RF increased during the leg swing initiation. RF and SEM can be considered as a pair of antagonistic muscles, and responses of antagonistic muscle pairs are related to external assistance.

**Table 1.** Statistical results of the kinematic data under four conditions.

| | | CON | EX003 | EX022 | EX041 | p-value |
|---|---|---|---|---|---|---|
| ROM | hip (degrees) | 51.2 ± 2.8 | 51.0 ± 2.5 | 51.2 ± 2.8 | 50.9 ± 2.1 | 0.72 |
| | knee (degrees) | 66.0 ± 6.4 | 64.3 ± 5.6 | 63.3 ± 4.9 | 62.2 ± 5.1 | <0.01 |
| | ankle (degrees) | 30.4 ± 4.0 | 30.0 ± 4.5 | 31.0 ± 4.1 | 30.4 ± 4.4 | 0.42 |
| gait parameters | SF (Hz) | 1.92 ± 0.13 | 1.95 ± 0.16 | 1.98 ± 0.14 | 2.01 ± 0.16 | <0.01 |
| | SL (m) | 0.69 ± 0.05 | 0.68 ± 0.05 | 0.67 ± 0.05 | 0.66 ± 0.05 | <0.01 |
| | SW (m) | 0.11 ± 0.03 | 0.12 ± 0.04 | 0.11 ± 0.03 | 0.11 ± 0.03 | 0.50 |
| gait parameters variability | SF variability ($10^{-2}$ Hz) | 3.59 ± 0.89 | 3.77 ± 1.28 | 3.41 ± 0.89 | 3.52 ± 1.45 | 0.65 |
| | SL variability ($10^{-2}$ m) | 1.47 ± 0.42 | 1.71 ± 0.57 | 1.38 ± 0.39 | 1.44 ± 0.49 | 0.21 |
| | SW variability ($10^{-2}$ m) | 1.70 ± 0.50 | 1.71 ± 0.55 | 1.60 ± 0.41 | 1.37 ± 0.38 | 0.03 |

One study proposed that the antagonistic muscle pairs may contract simultaneously when the force applied to the body exceeds a certain threshold, while both of them may have a lower activation when the force is lower than the threshold [43]. Nevertheless, it remains unclear why the activation of RF increased during the unassisted phase. One possible reason is related to the increased SF, which causes more rapid leg swing initiation, such that more muscle forces may be required. Notably, although the exotendon in this study provided assistance in hip extensors and knee flexors, GM only showed a decreasing trend, and GAS was almost not affected. This may be related to the main functions of these two muscles. GM mainly supports the body weight during the stance phase, and GAS mainly propels the body centre of mass during push-off. The timing and low magnitude of the assistance may limit the benefits to these two muscles.

The biological joint moment is the net moment of muscle force, which means that a lower burden of muscles is related to the reduction of biological joint moment. Different assistive devices with different assistance parameters may have a different effect on the joint moment [33,44–46]. In this study, the biological joint moment did show significant differences among conditions. These differences can be attributed to two aspects, i.e. the assistance provided by the external force and active adjustment made by the human body. As expected, the hip extension and knee flexion moments during the late swing phase and the early stance phase decreased with the exotendon assistance. Unexpectedly, the knee flexion moment during the late stance phase and the ankle plantarflexion moment also decreased when receiving no external assistance. The hip flexion moment during the middle and late stance phases showed an increasing trend. These changes may be due to the adjustments made by the human body, such as altered kinematics. Since the joint moments were calculated by inverse dynamics, altered kinematics indicates the change of inertial force. Adjustment for the joint moment will challenge the design of assistive devices, because assistance parameters, such as the timing and magnitude of the assistance, are usually determined based on the human joint moment, which implies an assumption that the joint moment remains unchanged when applying external assistive forces [23,43,47]. In our study, the active adjustment of humans did cause a significant change of joint moment during the unassisted phase, but on the whole the adjustment was positive.

To a certain extent, altered kinematics may bring uncertainties about the performance of assistive devices, as it can also induce deviations of the assistance parameters from the expected and cause discomfort due to deviations from normal gait. In this study, the kinematics was significantly changed, where the hip flexion, knee flexion and extension decreased, and the hip extension increased. The change increased with the assistance magnitude. Many previous studies also reported alterations in kinematics more or less [6,25,31,33,37,44]. The reasons reported seem insufficient to account for the difference in our study, such as the added mass [37,43,48] and movement constraints caused by structures [49]. Given that the assistance magnitude was far less than the moment required for walking in our study, theoretically, the provided assistance will not change the kinematics. We speculated that there were two possible reasons. One is that humans did not entirely adapt to walking with the exotendon, and longer familiarization was possibly needed. The other is that humans made adjustments actively to maximize efficiency, similar to findings from previous studies [28,33]. The latter is more persuasive, as we found the change also occurred in the unassisted phase, such as the increased hip extension during the late stance phase. This may be due to the compensation mechanism, where reduced hip flexion around heel-strike was compensated by the increased hip extension during push-off to minimize the change of hip ROM. The knee joint ROM decreased significantly, and the decrease may cause the decrease of SL and increase of SF, with SW almost unaffected. Gait variability involves the stability problem during walking [50], but it was seldom evaluated in previous studies. Only a recent study tested if there were tripping incidents in overground walking [33]. Actually, a sudden transition to a new gait possibly caused by the assistive device can easily bring about the stability problem [51]. In this study, the SF variability and the SL variability were not affected, and the SW variability was improved. The improvement may be related to the arrangement of the exotendon. Although it mainly provides forces in the sagittal plane, forces may also be produced in the frontal plane when SW changes, which may reduce the gait change in the frontal plane.

Notably, measurements that show significant changes varied monotonically with the stiffness. In particular, the assisted hamstring muscles decreased with the spring stiffness, which was consistent with previous studies where the assisted soleus decreased with the assistance magnitude using active and passive ankle exoskeleton [6,8]. In this study, the timings and magnitudes of the assistance for different spring stiffness conditions can be different, as the resting length of the exotendon was

determined individually for each spring stiffness. However, roughly higher stiffness caused higher assistance magnitude. Overall, the current study had a relatively low assistance magnitude, and the decision was based on two considerations. First, the dynamic stiffness of the hip and knee joints during the swing phase is smaller than that during the stance phase [52,53], which makes it easy to alter the joint kinematics. Second, because no rigid structures exist in our device, greater assistive forces mean that soft tissues have to bear more deformation, which may make subjects uncomfortable [54]. Actually, a higher magnitude does not always produce better performance [6,8,55]. Providing small but effective assistance is also one of the researchers' pursuits [32].

Admittedly, there are several limitations in this study. Firstly, the profile of the assistive torque did not resemble the human joint moment. Limited by the physical characteristics of the exotendon, the produced forces and the moment arm at a certain moment are difficult to adjust as expected. Also, to increase the moment arm, the lower attachment is located at the distal end of the shank, which causes a large variation of the moment arm during a gait cycle. Thus, the moment arm ratio of the hip to knee likewise exhibits significant variation. Inappropriate moment arm ratio possibly causes resistance to the joint movement. However, due to the low assistance magnitude in our study, the effect was limited. Secondly, the optimized parameters did not necessarily mean the optimal assistance, which was limited by the assumptions of the optimization. For example, we assumed that the muscle effort will increase with the total biological joint moment, and that the kinematics and kinetics remain unchanged. Additionally, the actual stiffness of the device is lower than the theoretical one due to the soft tissues. Nevertheless, the optimization was still of significance. Intuitively determining the physical parameters may make the result slightly more unpredictable. Optimization can provide a more specific understanding of the assistance strategy before commencing the experiment, thus helping us to estimate whether the parameters are appropriate.

## 5. Conclusion

In the current study, we presented a portable and lightweight exotendon across the hip and knee, and then we investigated the effects of the exotendon on the muscle effort, joint kinetics and kinematics during walking. With the assistance of the exotendon, the biological hip extension and knee flexion moments were significantly reduced. Activation of the assisted semitendinosus was lower, and activation of other measured muscles did not show significant differences. Certainly, the effectiveness of assistive devices is inseparable from appropriate physical parameters. Although the optimization aiming at minimizing the biological joint moments cannot ensure the optimal performance, it is of significance to avoid the need for intuitive determination of physical parameters. Future studies can determine these parameters using a more accurate musculoskeletal model, which may leave the optimization results more persuasive.

Ethics. This study was approved by the Chinese Ethics Committee of Registering Clinical Trials, ethical permit no. ChiECRCT20200228. All subjects participating in this study gave their written informed consent.
Data accessibility. The data are provided in the electronic supplementary material [56].
Authors' contributions. C.X. and L.C. designed the study. L.C. and B.H. built the device. L.C. and X.X. performed the experiments. L.C., C.X., W.C. and J.L. analysed and interpreted the results of the study. L.C. drafted the manuscript, and C.X., W.C., J.L., B.H. and X.X. revised the manuscript. All authors approved the final manuscript.
Competing interests. We declare we have no competing interests.
Funding. This work was partially supported by the National Natural Science Foundation of China (grant nos. 52027806, 52005191, U1911601, 52075191, 91648203 and U1913205) and the Hubei Provincial Natural Science Foundation (grant no. 2020CFB424).
Acknowledgements. We would like to thank Qinhao Zhang for his assistance in data collection.

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
