## [Peer Review File · Royal Society Open Science]

Review History

RSOS-201298.R0 (Original submission)

Review form: Reviewer 1

Is the manuscript scientifically sound in its present form?

Yes

Are the interpretations and conclusions justified by the results?

Yes

Is the language acceptable?

Yes

Do you have any ethical concerns with this paper?

No

Have you any concerns about statistical analyses in this paper?

No

Recommendation?

Accept as is

Comments to the Author(s)

The manuscript is ready for publication.

One note that the authors may or may not choose to address in this manuscript: The amount of time provided for training was far less than needed in other studies targeting subtle improvements with passive devices. For example, in both Simpson et al. and Collins et al., participants trained for four sessions, each including more than an hour of locomotion.

Review form: Reviewer 2

Is the manuscript scientifically sound in its present form?

No

Are the interpretations and conclusions justified by the results?

Yes

Is the language acceptable?

Yes

Do you have any ethical concerns with this paper?

No

Have you any concerns about statistical analyses in this paper?

No

Recommendation?

Major revision is needed (please make suggestions in comments)

Comments to the Author(s)

This paper presents a biarticular passive assistive device based on the concept of exotendon (introduced by van den Bogert). This passive device contributes to hip extension and knee flexion and supports human walking in the late swing and early stance phases. The results show a decrease in Semitendosus muscle without significant changes in other muscles as well as gait kinematic and kinetic behavior.

The paper is clear and presents a method for optimizing the device parameters (stiffness and rest length of the spring). However, the main contribution of the paper is not new. The same concept in a comparable design was introduced in a biarticular thigh exosuit in the following article [a]. There are several studies on biarticular muscles and the applications in robotics and assistive technologies including two review papers [b,c] and several other simulation and experimental studies which are ignored by the authors. The only difference between the design in the introduced exosuit in [a] and the device of this paper is that in [a] both two antagonistic biarticular muscles are used to mimic RF and HAM muscles, and this paper uses just HAM-like muscle. If we accept that this is a sufficient contribution for a new article, the authors at least need to refer to this former study and explain the differences.

In addition to this significant drawback, the design concept and the details are not solid. The authors neglected the importance of the ratio between hip to knee lever arm which is an important design concept. There are some incomplete argumentations and although I see the importance of the analyses to show the effects of the designed exotendon on supporting human gait, minor improvement in one muscle activation while neglecting some other important muscles which might be affected such as hip monoarticular muscles (e.g., gluteus maximus) is not justified. Further, the kinetic and kinematic analyses in separate experiments, unnecessary descriptions and unclear figures make this paper inappropriate to be accepted for publication in its current format. See more details in the following.

Detailed comments:

Introduction:

The authors nicely explained the literature on related passive assistive devices and motivated the attention to hip and knee joints. However, they ignored many relevant studies on biarticular muscle both biomechanics and assistive technologies. There are several articles on the design and control of active and passive exosuits using biarticular thigh muscle design (see some references below).

The main contribution in this paper is the biarticular design which has several advantages. The authors did not sufficiently motivate this important design feature. It seems that the only reason for designing the device is that the others did not do it which is not correct, as explained above. I encourage the authors to have a better literature review about the role of biarticular muscles in biomechanical and robotic studies to better motivate developing such a device. The following papers could be helpful.

Methods:

The authors focused on the stiffness and rest length of the artificial muscles (spring), while the hip to knee lever arm ratio plays an important role in specifying the contribution at each joint. In the proposed design, there is no methodology to define the lever arms. Selecting the bottom attachment point around the ankle results in a large variation in the knee lever arm which is not suitable for replicating human joint torques.

The authors assume that the muscle efforts will be decreased which should be verified at the end. However, this is not supported as the hip monoarticular muscles' activations are not measured. Maybe there is an increase in hip flexor muscles (not measured) as the spring acts against it.

In the second session of the preliminary experiments on the treadmill, the subjects wear the exotendon and the stiffness considered in these experiments will affect the movement and then the force direction. It is not described how the stiffness and rest length in these experiments are selected. For example, if the springs are too soft or too stiff, the results will be different. Further, doing separate experiments on the treadmill and walkway for measuring the kinematics and kinetics is not justified. We know that treadmill walking and ground walking are not similar. In addition, inscribing step frequency with a metronome does not mean similar speeds as the step length might be different.

The description of the joint torque calculation is too long with unnecessary formulations in appendix 1. All equations are trivial and not necessary. For example the weird term "and the assistive torque produced by one Newton force that was calculated from the preliminary experiment." can be replaced by "and the lever arm".

Equation 2.4 is used for optimization, but in the results, it is not used for evaluating the quality of the device in decreasing the muscle contribution. It seems that Fig. 5.b and Fig. 7 show the total moment, which is the summation of the human muscle and the springs as calculated by the GRF through inverse dynamics.

It seems that the maximum stiffness in the range is found as the optimal value. Why? can you explain what happens? What if the range is increased. The equation can be solved analytically to find the global minimum and the optimization result can be found with a closed-form equation – no need for GA optimization to find the local minimum in the range, although it is acceptable.

The following sentence is not precise as it will move forward again at the next push-off:
 "The gait cycle was segmented by the marker on the right heel, where heel-strike was thought to occur when the marker moved to the anterior most position."

To calculate the total joint moment, why don't you use the integral of the absolute of the joint moment (similar to 2.4) as the negative and positive torques might compensate each other without using absolute.

Results:

The figures at the end of the paper are all corrupted. All the texts are mixed with figures and the line numbers, and legends.

The authors mentioned the discrepancy between the timing of the Hip and Knee peak torques prior to heel-strike (Fig.2). This is different from the human torque patterns (Fig. 5b) which might result from the lever arm adjustment.

Which one is correct? Scaling factor or scalar factor?

It is stated that "the maximum extension of the hip increased by 6.9 degrees", but this is not visible in the figure. The difference between EXO and CON experiments in hip flexion (negative peak) is more pronounced (in both angle and moment figures) than the slightly reduced extension peak explained by the authors. What is the reason for this increase flexion torque in the region that the exo does not contribute?

Figs 8-10 can be removed. Not informative! Fig. 10 is not even described. What is the orange line. It is not readable and it is not clear why the authors show it.

Discussion

The goal of the design was to minimize the total biological joint moment of the hip and knee, as mentioned also at the beginning of the discussion. However, it was not supported by the results and is not discussed in the discussions.

Decreasing the interface torque over time is expected due to the method of attaching the two sides of the exotendon to the body. This is a known phenomenon and it is not worth to be part of the results and also be discussed in detail in Sec. 4. There are also methods to prevent it, such as using a strap below the shoe sole to prevent the lower segment movement.

When the authors discussed other options, they argue "[34]. Actually, larger magnitude does not always produce better performance [7, 18, 35]." This is correct, but it does not mean that we should not examine higher magnitudes because it might provide inconvenience. Further, the lever arm effects should be analysed more carefully and not just the magnitude.

When the authors discuss the effects on other hip extensors and knee flexor muscles, they explain gastrocnemius, but not Gluteus Maximus, which was not even measured. This is one of the important muscles which should be analyzed.

The following statement is also not completely accurate as in the first 10% of the stance phase, the spring supports knee torque:

"i.e., the exotendon will hinder the knee joint movement during the late period of the early stance phase. However"

In the following sentence the authors discuss the knee flexion. What about the hip flexion (peak) which shows more pronounced difference.

"Similarly, the knee flexion decreased during the unassisted phase, which may be related to the necessity of altering the ground clearance due to change in the step length."

The following sentence is not correct as gait variation and stability are two different concepts. An assistive device might change the gait and make it even more stable.

"In addition, in this study the gait variability did not show a significant difference between conditions, which suggested that the exotendon did not introduce the stability problems."

Later the authors say, ". In our study the lightweight design and small assistance may account for the unaffected stability." This is also not strong as the same device with too stiff spring might result in a different outcome.

Useful References to be added:

[a] Barazesh et al, A biarticular passive exosuit to support balance control can reduce metabolic cost of walking, *Bioinspiration, Biomimetics*, 2020.

[b] Schumacher et al., Biarticular muscles in light of template models, experiments and robotics: a review, *the Royal Society Interface*, 2020.

[c] Junius et al., Biarticular elements as a contributor to energy efficiency: biomechanical review and application in bio-inspired robotics. *Bioinspiration Biomimetics*, 2017

[d] Hosoda et al, Actuation in legged locomotion.

In *Bioinspired legged locomotion*, pp. 563–622. Elsevier, 2017.

[e] Hof. The force resulting from the action of mono- and biarticular muscles in a limb. *Journal of Biomechanics*. 2001

[f] Sharbafi et al., Reconstruction of human swing leg motion with passive biarticular muscle models. *Human Movement Science*, 2017.

[g] Schumacher, et al., Biarticular muscles are most responsive to upper-body pitch perturbations in human standing. *Scientific reports*, 2019.

[h] Sharbafi et al., Leg force control through biarticular muscles for human walking assistance. *Frontiers in Neurorobotics*, 2018.

[i] Németh et al., In vivo moment arm lengths for hip extensor muscles at different angles of hip flexion. *Journal of Biomechanics*, 1985.

[j] Visser et al., Length and moment arm of human leg muscles as a function of knee and hip-joint angles. *Eur. J. Appl. Physiol. Occup.* 1990.

[k] van Dijk, et al, A Passive Exoskeleton with Artificial Tendons, *ICORR* 2011.

[l] Eilenberg et al., Development and evaluation of a powered artificial gastrocnemius for transtibial amputee gait. *Journal of Robotics*. 2018,

[m] Quinlivan et al. Assistance magnitude versus metabolic cost reductions for a tethered multiarticular soft exosuit. *Sci. Robot.* 2017.

[n] Schmidt et al., The Myosuit: bi-articular anti-gravity exosuit that reduces hip extensor activity in sitting transfers. *Frontiers in Neurorobotics*, 2017

Decision letter (RSOS-201298.R0)

Dear Mr Cheng

The Editors assigned to your paper RSOS-201298 "A portable exotendon assisting hip and knee joints reduces muscular burden during walking" have now received comments from reviewers and would like you to revise the paper in accordance with the reviewer comments and any comments from the Editors. Please note this decision does not guarantee eventual acceptance.

Please submit your revised manuscript and required files (see below) no later than 21 days from today's (ie 07-Jan-2021) date. Note: the ScholarOne system will 'lock' if submission of the revision is attempted 21 or more days after the deadline. If you do not think you will be able to meet this deadline please contact the editorial office immediately.

on behalf of Prof R. Kerry Rowe (Subject Editor)
openscience@royalsociety.org

Associate Editor Comments to Author:

Thank you for your patience while we sought review of this work - unfortunately, the original reviewers at JRSI were not available for this version of the paper. Given the comments of the referees we have received reports from, however, we would like you to revise the paper to take into account the reviewers' concerns. The more critical reviewer has recommended a number of

additional references to include -- as it is not clear from the referee's comments where and how all these references are relevant to your work (except in a generalised sense), we do not expect you to add them all without thinking. Please ensure that you only include specific references that support specific aspects of your paper - adding citations simply for the sake of adding them helps no one, and reinforces the skewing of and overreliance on journal impact factors as a metric, without adding any value to the literature.

Reviewer comments to Author:

Reviewer: 1

Comments to the Author(s)

The manuscript is ready for publication.

One note that the authors may or may not choose to address in this manuscript: The amount of time provided for training was far less than needed in other studies targeting subtle improvements with passive devices. For example, in both Simpson et al. and Collins et al., participants trained for four sessions, each including more than an hour of locomotion.

Reviewer: 2

Comments to the Author(s)

This paper presents a biarticular passive assistive device based on the concept of exotendon (introduced by van den Bogert). This passive device contributes to hip extension and knee flexion and supports human walking in the late swing and early stance phases. The results show a decrease in Semitendosus muscle without significant changes in other muscles as well as gait kinematic and kinetic behavior.

The paper is clear and presents a method for optimizing the device parameters (stiffness and rest length of the spring). However, the main contribution of the paper is not new. The same concept in a comparable design was introduced in a biarticular thigh exosuit in the following article [a]. There are several studies on biarticular muscles and the applications in robotics and assistive technologies including two review papers [b,c] and several other simulation and experimental studies which are ignored by the authors. The only difference between the design in the introduced exosuit in [a] and the device of this paper is that in [a] both two antagonistic biarticular muscles are used to mimic RF and HAM muscles, and this paper uses just HAM-like muscle. If we accept that this is a sufficient contribution for a new article, the authors at least need to refer to this former study and explain the differences.

In addition to this significant drawback, the design concept and the details are not solid. The authors neglected the importance of the ratio between hip to knee lever arm which is an important design concept. There are some incomplete argumentations and although I see the importance of the analyses to show the effects of the designed exotendon on supporting human gait, minor improvement in one muscle activation while neglecting some other important muscles which might be affected such as hip monoarticular muscles (e.g., gluteus maximus) is not justified. Further, the kinetic and kinematic analyses in separate experiments, unnecessary descriptions and unclear figures make this paper inappropriate to be accepted for publication in its current format. See more details in the following.

Detailed comments:

Introduction:

The authors nicely explained the literature on related passive assistive devices and motivated the attention to hip and knee joints. However, they ignored many relevant studies on biarticular muscle both biomechanics and assistive technologies. There are several articles on the design and

control of active and passive exosuits using biarticular thigh muscle design (see some references below).

The main contribution in this paper is the biarticular design which has several advantages. The authors did not sufficiently motivate this important design feature. It seems that the only reason for designing the device is that the others did not do it which is not correct, as explained above. I encourage the authors to have a better literature review about the role of biarticular muscles in biomechanical and robotic studies to better motivate developing such a device. The following papers could be helpful.

Methods:

The authors focused on the stiffness and rest length of the artificial muscles (spring), while the hip to knee lever arm ratio plays an important role in specifying the contribution at each joint. In the proposed design, there is no methodology to define the lever arms. Selecting the bottom attachment point around the ankle results in a large variation in the knee lever arm which is not suitable for replicating human joint torques.

The authors assume that the muscle efforts will be decreased which should be verified at the end. However, this is not supported as the hip monoarticular muscles' activations are not measured. Maybe there is an increase in hip flexor muscles (not measured) as the spring acts against it.

In the second session of the preliminary experiments on the treadmill, the subjects wear the exotendon and the stiffness considered in these experiments will affect the movement and then the force direction. It is not described how the stiffness and rest length in these experiments are selected. For example, if the springs are too soft or too stiff, the results will be different. Further, doing separate experiments on the treadmill and walkway for measuring the kinematics and kinetics is not justified. We know that treadmill walking and ground walking are not similar. In addition, inscribing step frequency with a metronome does not mean similar speed as the step length might be different.

The description of the joint torque calculation is too long with unnecessary formulations in appendix 1. All equations are trivial and not necessary. For example the weird term "and the assistive torque produced by one Newton force that was calculated from the preliminary experiment." can be replaced by "and the lever arm".

Equation 2.4 is used for optimization, but in the results, it is not used for evaluating the quality of the device in decreasing the muscle contribution. It seems that Fig. 5.b and Fig. 7 show the total moment, which is the summation of the human muscle and the springs as calculated by the GRF through inverse dynamics.

It seems that the maximum stiffness in the range is found as the optimal value. Why? can you explain what happens? What if the range is increased. The equation can be solved analytically to find the global minimum and the optimization result can be found with a closed-form equation — no need for GA optimization to find the local minimum in the range, although it is acceptable.

The following sentence is not precise as it will move forward again at the next push-off:
 "The gait cycle was segmented by the marker on the right heel, where heel-strike was thought to occur when the marker moved to the anterior most position."

To calculate the total joint moment, why don't you use the integral of the absolute of the joint moment (similar to 2.4) as the negative and positive torques might compensate each other without using absolute.

Results:

The figures at the end of the paper are all corrupted. All the texts are mixed with figures and the line numbers, and legends.

The authors mentioned the discrepancy between the timing of the Hip and Knee peak torques prior to heel-strike (Fig.2). This is different from the human torque patterns (Fig. 5b) which might result from the lever arm adjustment.

Which one is correct? Scaling factor or scalar factor?

It is stated that "the maximum extension of the hip increased by 6.9 degrees", but this is not visible in the figure. The difference between EXO and CON experiments in hip flexion (negative peak) is more pronounced (in both angle and moment figures) than the slightly reduced extension peak explained by the authors. What is the reason for this increase flexion torque in the region that the exo does not contribute?

Figs 8-10 can be removed. Not informative! Fig. 10 is not even described. What is the orange line. It is not readable and it is not clear why the authors show it.

Discussion

The goal of the design was to minimize the total biological joint moment of the hip and knee, as mentioned also at the beginning of the discussion. However, it was not supported by the results and is not discussed in the discussions.

Decreasing the interface torque over time is expected due to the method of attaching the two sides of the exotendon to the body. This is a known phenomenon and it is not worth to be part of the results and also be discussed in detail in Sec. 4. There are also methods to prevent it, such as using a strap below the shoe sole to prevent the lower segment movement.

When the authors discussed other options, they argue "[34]. Actually, larger magnitude does not always produce better performance [7, 18, 35]." This is correct, but it does not mean that we should not examine higher magnitudes because it might provide inconvenience. Further, the lever arm effects should be analysed more carefully and not just the magnitude.

When the authors discuss the effects on other hip extensors and knee flexor muscles, they explain gastrocnemius, but not Gluteus Maximus, which was not even measured. This is one of the important muscles which should be analyzed.

The following statement is also not completely accurate as in the first 10% of the stance phase, the spring supports knee torque:

"i.e., the exotendon will hinder the knee joint movement during the late period of the early stance phase. However"

In the following sentence the authors discuss the knee flexion. What about the hip flexion (peak) which shows more pronounced difference.

"Similarly, the knee flexion decreased during the unassisted phase, which may be related to the necessity of altering the ground clearance due to change in the step length."

The following sentence is not correct as gait variation and stability are two different concepts. An assistive device might change the gait and make it even more stable.

"In addition, in this study the gait variability did not show a significant difference between conditions, which suggested that the exotendon did not introduce the stability problems."

Later the authors say, ". In our study the lightweight design and small assistance may account for the unaffected stability." This is also not strong as the same device with too stiff spring might result in a different outcome.

Useful References to be added:

- [a] Barazesh et al, A biarticular passive exosuit to support balance control can reduce metabolic cost of walking, *Bioinspiration, Biomimetics*, 2020.
- [b] Schumacher et al., Biarticular muscles in light of template models, experiments and robotics: a review, *the Royal Society Interface*, 2020.
- [c] Junius et al., Biarticular elements as a contributor to energy efficiency: biomechanical review and application in bio-inspired robotics. *Bioinspiration Biomimetics*, 2017
- [d] Hosoda et al, Actuation in legged locomotion. In *Bioinspired legged locomotion*, pp. 563–622. Elsevier, 2017.
- [e] Hof. The force resulting from the action of mono- and biarticular muscles in a limb. *Journal of Biomechanics*. 2001
- [f] Sharbafi et al., Reconstruction of human swing leg motion with passive biarticular muscle models. *Human Movement Science*, 2017.
- [g] Schumacher, et al., Biarticular muscles are most responsive to upper-body pitch perturbations in human standing. *Scientific reports*, 2019.
- [h] Sharbafi et al., Leg force control through biarticular muscles for human walking assistance. *Frontiers in Neurorobotics*, 2018.
- [i] Németh et al., In vivo moment arm lengths for hip extensor muscles at different angles of hip flexion. *Journal of Biomechanics*, 1985.
- [j] Visser et al., Length and moment arm of human leg muscles as a function of knee and hip-joint angles. *Eur. J. Appl. Physiol. Occup.* 1990.
- [k] van Dijk, et al, A Passive Exoskeleton with Artificial Tendons, *ICORR* 2011.
- [l] Eilenberg et al., Development and evaluation of a powered artificial gastrocnemius for transtibial amputee gait. *Journal of Robotics*. 2018,
- [m] Quinlivan et al. Assistance magnitude versus metabolic cost reductions for a tethered multiarticular soft exosuit. *Sci. Robot*. 2017.
- [n] Schmidt et al., The Myosuit: bi-articular anti-gravity exosuit that reduces hip extensor activity in sitting transfers. *Frontiers in Neurorobotics*, 2017

===PREPARING YOUR MANUSCRIPT===

Please ensure that you include an acknowledgements' section before your reference list/bibliography. This should acknowledge anyone who assisted with your work, but does not

qualify as an author per the guidelines at <https://royalsociety.org/journals/ethics-policies/openness/>.

===PREPARING YOUR REVISION IN SCHOLARONE===

<https://royalsociety.org/journals/authors/author-guidelines/#supplementary-material> to include a suitable title and informative caption. An example of appropriate titling and captioning may be found at https://figshare.com/articles/Table_S2_from_Is_there_a_trade-off_between_peak_performance_and_performance_breadth_across_temperatures_for_aerobic_scorpions_in_teleost_fishes_/3843624.

Author's Response to Decision Letter for (RSOS-201298.R0)

See Appendix A.

RSOS-201298.R1 (Revision)

Review form: Reviewer 2

Is the manuscript scientifically sound in its present form?

No

Are the interpretations and conclusions justified by the results?

No

Is the language acceptable?

Yes

Do you have any ethical concerns with this paper?

No

Have you any concerns about statistical analyses in this paper?

No

Recommendation?

Major revision is needed (please make suggestions in comments)

Comments to the Author(s)

The revised paper is improved in many aspects. However, the authors could not tackle all the significant drawbacks of the paper which were clearly described in my previous review. I like to give the authors another chance to revise the paper while the paper in its current condition is still not appropriate for publication.

● Major concern regarding the contributions:

■ In the previous review I mentioned that there are several relevant studies that should be read and considered to clarify the contribution of this work to the state-of-the-art. Although few new articles are cited in the revised version, it is done in a very superficial way. The key concept of the design of the exo which is using a biarticular spring is not motivated and is not linked to other studies. The authors described the contributions of this study compared to [22] which use the same design concept in the response letter, but in the paper adding two neutral sentences seems to be an attempt to hide this close relation between these two studies. I believe if the authors honestly describe the differences to the state-of-the-art, it elates the quality of the paper and not lowers your work. PLEASE carefully place your work in the stat-of-the-art.

■ The motivation of the work and the hypothesis is still not clear. Why do you use biarticular springs? After writing several paragraphs about passive and active exos, the last paragraph of into says "In this study, we aimed at further investigating the biarticular assistance strategy and examining the effects of the strategy on human movements". Why biarticular? what is the hypothesis behind it?

■ The authors referred to the work from van dijk 2014 [20] which was another implementation of the exotendon. You correctly mentioned that "The biological joint moment significantly decreases.", but you did not mention that the conclusion of [20] in the invalidity of the exotendon concept as the metabolic cost is not reduced. It seems that one of the main findings in your paper is also the reduction of biological torque which is also not supported by your outcomes (see below for detail). If we accept that you can reduce biological joint moments, it is still not sufficient to support the effectiveness of the exo.

3- In the current format of the paper, it is more like a technical report which describes the details of the experiments and presents the outcomes with no clear motivation, hypothesis, justification, or comprehensive discussion about the findings. There is also no clear key message

● Detailed comments:

Many of the comments from previous review were nicely taken into account in this revision. However, there are still some remaining issues. Here I use the numbering which was utilized in the response letter to facilitate communication:

■ comment 6: The lever arm is one parameter but a crucial one that can change the ratio of the generated torque at hip and knee and it is not reasonable to not discuss it because it is one parameter. The response of the authors is not convincing at all.

■ comment 7: The authors just responded to the previous comments superficially by saying this is the limitation of our study and did not modify the paper accordingly they even did not describe them carefully in the discussion. The authors should know that the goal of criticizing the paper is not to convince the reviewer, but to improve the quality of the paper which is not changed (except adding two references and few neutral sentences in the intro) from the previous version.

comment 8: Instead of addressing the comments the authors just repeated what was written in the paper in the response letter! The way of calculating the lever arm was already described in

the paper. The problem is that it was not considered in the design and after doing the experiments it was not considered in explaining the findings.

Again the response letter says "... But indeed it cannot ensure it. This measurement may affect the optimization results, but we thought that the effect is limited and acceptable." This must be discussed in the paper, not in the response letter.

Comment 10: The argumentation about decreasing biological joint torque is not correct as in some conditions the Exo torque might oppose the biological torque. This means even with similar total joint torque the biological torque could increase with exo. If we accept the argumentation from the authors, even if we use a very stiff spring or damper and ask the subjects to keep the same kinematic behavior, and the GRF stays the same, the biological torque will reduce which we know it is not correct. As t is one of the key outcomes of the paper it needs to be scientifically supported.

Comment 11: I understand the argumentation about partial support of the joint torques. However it is not emergent true to say if the range of stiffness is not limited, the optimized assistive torque will be comparable to human joint torque because you are trying to approach both hip and knee torque just by one biarticular passive spring. Since the knee and hip torques are reciprocal just in part of the gait cycle optimization of HAM-like stiffness should not necessarily give similar joint moments. If we accept this reasoning why do you need optimization?

Comment 12: In this sentence, it is not clear that this method is used for treadmill walking. Moreover, the leg moves backward prior to heelstrike which is called swing leg retraction. This means that the most anterior position of the heel marker is not exactly the heel strike.

Comment 19: As I explained above the argumentation about reduced biological torque is not correct. Even if you show it, still the exo can be unhelpful as shown in the study from van dijk. Then, what we can learn from this study?

Comment 21: What is presented in the discussion does not address the comment. The main point is not about having constant or variable lever arms. The authors should try to approach human-like lever arm ratio to generate consistent joint torques to humans. Even if it is variable, simple solutions like finding appropriate attachment points, mechanisms like cam or guides can help provide similar hip to knee lever arm $\langle u \rangle \text{RATIO} \langle /u \rangle$ which is more important than the value of the lever arm. If the ratio is too different from what observed in the human body, the contribution of the exo will contradict human muscle actuation. The authors need to discuss this effect instead of saying we use a variable lever arm because we cannot keep the lever arm fixed!

Comment 25: It is clear what gait variability means. The stability term is not clear in your argumentation. Stability has a clear meaning in control engineering and gait analyses. If you mean gait stability (repetitive movement without falling like a stable limit cycle) it cannot be identified by the variability between different experimental scenarios but within one specific experiment. If you mean the similarity to the normal gait you can argue like this, but mainly the first meaning is realized from stability.

Decision letter (RSOS-201298.R1)

Dear Mr Cheng

The Editors assigned to your paper RSOS-201298.R1 "A portable exotendon assisting hip and knee joints reduces muscular burden during walking" have made a decision based on their reading of the paper and any comments received from reviewers.

Regrettably, in view of the reports received, the manuscript has been rejected in its current form. However, a new manuscript may be submitted which takes into consideration these comments.

We invite you to respond to the comments supplied below and prepare a resubmission of your manuscript. Below the referees' and Editors' comments (where applicable) we provide additional requirements. We provide guidance below to help you prepare your revision.

Please note that resubmitting your manuscript does not guarantee eventual acceptance, and we do not generally allow multiple rounds of revision and resubmission, so we urge you to make every effort to fully address all of the comments at this stage. If deemed necessary by the Editors, your manuscript will be sent back to one or more of the original reviewers for assessment. If the original reviewers are not available, we may invite new reviewers.

Please resubmit your revised manuscript and required files (see below) no later than 06-Sep-2021. Note: the ScholarOne system will 'lock' if resubmission is attempted on or after this deadline. If you do not think you will be able to meet this deadline, please contact the editorial office immediately.

Please note article processing charges apply to papers accepted for publication in Royal Society Open Science (<https://royalsocietypublishing.org/rsos/charges>). Charges will also apply to papers transferred to the journal from other Royal Society Publishing journals, as well as papers submitted as part of our collaboration with the Royal Society of Chemistry (<https://royalsocietypublishing.org/rsos/chemistry>). Fee waivers are available but must be requested when you submit your manuscript (<https://royalsocietypublishing.org/rsos/waivers>).

Thank you for submitting your manuscript to Royal Society Open Science and we look forward to receiving your resubmission. If you have any questions at all, please do not hesitate to get in touch.

on behalf of Professor R. Kerry Rowe (Subject Editor)
openscience@royalsociety.org

Associate Editor Comments to Author:

Thank you for supplying this revision. While you have made efforts to improve the paper, it seems that substantial concerns remain. Regrettably, Royal Society Open Science is not generally able to offer multiple rounds of revision - rather, it is expected that, unless there are good reasons it is not possible, authors will closely and comprehensively respond to reviewer and editor

comments. Unfortunately, a number of the comments from the reviewer suggest that much more work is needed with your submission, and so we must reject the paper in its current form. Nevertheless, as the reviewer has offered helpful commentary and suggestions for improvements (for which the editors are grateful), if you wish to comprehensively revise the paper and resubmit it, we would be willing to consider the manuscript - if, however, the reviewer (or reviewers if needed) remain unsatisfied by the resubmission, no further options to revise will be available with the journal.

Reviewer comments to Author:

Reviewer: 2

Comments to the Author(s)

The revised paper is improved in many aspects. However, the authors could not tackle all the significant drawbacks of the paper which were clearly described in my previous review. I like to give the authors another chance to revise the paper while the paper in its current condition is still not appropriate for publication.

● **Major concern regarding the contributions:**

■ In the previous review I mentioned that there are several relevant studies that should be read and considered to clarify the contribution of this work to the state-of-the-art. Although few new articles are cited in the revised version, it is done in a very superficial way. The key concept of the design of the exo which is using a biarticular spring is not motivated and is not linked to other studies. The authors described the contributions of this study compared to [22] which use the same design concept in the response letter, but in the paper adding two neutral sentences seems to be an attempt to hide this close relation between these two studies. I believe if the authors honestly describe the differences to the state-of-the-art, it elates the quality of the paper and not lowers your work. PLEASE carefully place your work in the stat-of-the-art.

■ The motivation of the work and the hypothesis is still not clear. Why do you use biarticular springs? After writing several paragraphs about passive and active exos, the last paragraph of into says "In this study, we aimed at further investigating the biarticular assistance strategy and examining the effects of the strategy on human movements". Why biarticular? what is the hypothesis behind it?

■ The authors referred to the work from van dijk 2014 [20] which was another implementation of the exotendon. You correctly mentioned that "The biological joint moment significantly decreases.", but you did not mention that the conclusion of [20] in the invalidity of the exotendon concept as the metabolic cost is not reduced. It seems that one of the main findings in your paper is also the reduction of biological torque which is also not supported by your outcomes (see below for detail). If we accept that you can reduce biological joint moments, it is still not sufficient to support the effectiveness of the exo.

3- In the current format of the paper, it is more like a technical report which describes the details of the experiments and presents the outcomes with no clear motivation, hypothesis, justification, or comprehensive discussion about the findings. There is also no clear key message

● **Detailed comments:**

Many of the comments from previous review were nicely taken into account in this revision. However, there are still some remaining issues. Here I use the numbering which was utilized in the response letter to facilitate communication:

■ comment 6: The lever arm is one parameter but a crucial one that can change the ratio of the generated torque at hip and knee and it is not reasonable to not discuss it because it is one parameter. The response of the authors is not convincing at all.

■ comment 7: The authors just responded to the previous comments superficially by saying this is the limitation of our study and did not modify the paper accordingly they even did not describe them carefully in the discussion. The authors should know that the goal of criticizing the paper is not to convince the reviewer, but to improve the quality of the paper which is not changed (except adding two references and few neutral sentences in the intro) from the previous version.

comment 8: Instead of addressing the comments the authors just repeated what was written in the paper in the response letter! The way of calculating the lever arm was already described in the paper. The problem is that it was not considered in the design and after doing the experiments it was not considered in explaining the findings.

Again the response letter says "... But indeed it cannot ensure it. This measurement may affect the optimization results, but we thought that the effect is limited and acceptable." This must be discussed in the paper, not in the response letter.

Comment 10: The argumentation about decreasing biological joint torque is not correct as in some conditions the Exo torque might oppose the biological torque. This means even with similar total joint torque the biological torque could increase with exo. If we accept the argumentation from the authors, even if we use a very stiff spring or damper and ask the subjects to keep the same kinematic behavior, and the GRF stays the same, the biological torque will reduce which we know it is not correct. As t is one of the key outcomes of the paper it needs to be scientifically supported.

Comment 11: I understand the argumentation about partial support of the joint torques. However it is not emergent true to say if the range of stiffness is not limited, the optimized assistive torque will be comparable to human joint torque because you are trying to approach both hip and knee torque just by one biarticular passive spring. Since the knee and hip torques are reciprocal just in part of the gait cycle optimization of HAM-like stiffness should not necessarily give similar joint moments. If we accept this reasoning why do you need optimization?

Comment 12: In this sentence, it is not clear that this method is used for treadmill walking. Moreover, the leg moves backward prior to heelstrike which is called swing leg retraction. This means that the most anterior position of the heel marker is not exactly the heel strike.

Comment 19: As I explained above the argumentation about reduced biological torque is not correct. Even if you show it, still the exo can be unhelpful as shown in the study from van dijk. Then, what we can learn from this study?

Comment 21: What is presented in the discussion does not address the comment. The main point is not about having constant or variable lever arms. The authors should try to approach human-like lever arm ratio to generate consistent joint torques to humans. Even if it is variable, simple solutions like finding appropriate attachment points, mechanisms like cam or guides can help provide similar hip to knee lever arm RATIO which is more important than the value of the lever arm. If the ratio is too different from what observed in the human body, the contribution of the exo will contradict human muscle actuation. The authors need to discuss this effect instead of saying we use a variable lever arm because we cannot keep the lever arm fixed!

Comment 25: It is clear what gait variability means. The stability term is not clear in your argumentation. Stability has a clear meaning in control engineering and gait analyses. If you

mean gait stability (repetitive movement without falling like a stable limit cycle) it cannot be identified by the variability between different experimental scenarios but within one specific experiment. If you mean the similarity to the normal gait you can argue like this, but mainly the first meaning is realized from stability.

===PREPARING YOUR MANUSCRIPT===

===PREPARING YOUR REVISION IN SCHOLARONE===

<https://royalsociety.org/journals/authors/author-guidelines/#supplementary-material> to include a suitable title and informative caption. An example of appropriate titling and captioning may be found at https://figshare.com/articles/Table_S2_from_Is_there_a_trade-off_between_peak_performance_and_performance_breadth_across_temperatures_for_aerobic_scops_in_teleost_fishes_/3843624.

Author's Response to Decision Letter for (RSOS-201298.R1)

See Appendix B.

RSOS-211266.R0

Review form: Reviewer 2

Is the manuscript scientifically sound in its present form?

Yes

Are the interpretations and conclusions justified by the results?

Yes

Is the language acceptable?

Yes

Do you have any ethical concerns with this paper?

No

Have you any concerns about statistical analyses in this paper?

No

Recommendation?

Accept with minor revision (please list in comments)

Comments to the Author(s)

The quality of the paper is significantly improved. The authors spent quite an effort to redo some experiments and also revise the paper to address all concerns. With the current version, I do not have any major concerns, but I found few minor issues (mentioned below). I would suggest revising the paper by a native speaker.

1- After the revision, the consistency of the paper needs to be checked. It seems that the authors rush in revising some parts which could be improved. For example, this sentence seems isolated and not well presented. "Secondly, assistance parameters are well selected."

2- I could say that the writing quality is lowered in some cases. Again, revising the paper by a native speaker is suggested. Here are some examples (not limited to that can be improved in the introduction section, but they are not limited to these points.

a) A biarticular knee-ankle exoskeleton has been found a higher reduction in metabolic cost than the monoarticular assistance.

b) More recently, exploring tasks of not too much effort is also popular, such as swing leg deceleration.

3- The rest length adjustment can be described and discussed more

Decision letter (RSOS-211266.R0)

Dear Mr Cheng

On behalf of the Editors, we are pleased to inform you that your Manuscript RSOS-211266 "A portable exotendon assisting hip and knee joints reduces muscular burden during walking" has been accepted for publication in Royal Society Open Science subject to minor revision in accordance with the referees' reports. Please find the referees' comments along with any feedback from the Editors below my signature.

Please submit your revised manuscript and required files (see below) no later than 7 days from today's (ie 01-Oct-2021) date. Note: the ScholarOne system will 'lock' if submission of the revision is attempted 7 or more days after the deadline. If you do not think you will be able to meet this deadline please contact the editorial office immediately.

on behalf of R. Kerry Rowe (Subject Editor)
openscience@royalsociety.org

Associate Editor Comments to Author:
Comments to the Author:

Thank you for this resubmission - it is clear you've put a lot of effort into preparing it. The reviewer (who re-reviewed) of your work has only a handful of relatively minor comments that you should address in a final revision. Please note that they have recommended you seek advice from a language editing service to ensure that the paper's message is not obscured by non-standard language forms. A number of services you might choose to use exist at <https://royalsociety.org/journals/authors/benefits/language-editing/>. Good luck and we'll look forward to receiving the revision.

Reviewer comments to Author:
Reviewer: 2

Comments to the Author(s)

The quality of the paper is significantly improved. The authors spent quite an effort to redo some experiments and also revise the paper to address all concerns. With the current version, I do not have any major concerns, but I found few minor issues (mentioned below). I would suggest revising the paper by a native speaker.

1- After the revision, the consistency of the paper needs to be checked. It seems that the authors rush in revising some parts which could be improved. For example, this sentence seems isolated and not well presented. "Secondly, assistance parameters are well selected."

2- I could say that the writing quality is lowered in some cases. Again, revising the paper by a native speaker is suggested. Here are some examples (not limited to that can be improved in the introduction section, but they are not limited to these points.

- a) A biarticular knee-ankle exoskeleton has been found a higher reduction in metabolic cost than the monoarticular assistance.
- b) More recently, exploring tasks of not too much effort is also popular, such as swing leg deceleration.

3- The rest length adjustment can be described and discussed more

===PREPARING YOUR MANUSCRIPT===

===PREPARING YOUR REVISION IN SCHOLARONE===

-- Ensure that your data access statement meets the requirements at <https://royalsociety.org/journals/authors/author-guidelines/#data>. You should ensure that you cite the dataset in your reference list. If you have deposited data etc in the Dryad repository, please only include the 'For publication' link at this stage. You should remove the 'For review' link.

Author's Response to Decision Letter for (RSOS-211266.R0)

See Appendix C.

Decision letter (RSOS-211266.R1)

Dear Mr Cheng,

I am pleased to inform you that your manuscript entitled "A portable exotendon assisting hip and knee joints reduces muscular burden during walking" is now accepted for publication in Royal Society Open Science.

on behalf of Prof R. Kerry Rowe (Subject Editor)

Appendix A

Dear editors and reviewers,

Thanks for giving us the opportunity to submit our revised manuscript entitled “A portable exotendon assisting hip and knee joints reduces muscular burden during walking” to the journal “Royal Society Open Science” for your reconsideration of its suitability for publication. We appreciate the time and effort that you dedicated to providing feedback on our manuscript. The comments provide valuable insights to refine the contents and analysis of our manuscript. Now, we have revised our manuscript according to the reviewers’ comments. All changes in the manuscript are marked in blue. Please see below, in blue, for a point-by-point response to the reviewers’ comments and concerns.

Yours sincerely,

Caihua Xiong

Institute of Robotics Research, Huazhong University of Science and Technology, Wuhan, Hubei 430074,

China

E-mail: chxiong@hust.edu.cn

Responses to Reviewer #1

General comment:

1. **Comment:** The manuscript is ready for publication.

Response: Thanks for your recognition.

2. **Comment:** One note that the authors may or may not choose to address in this manuscript: The amount of time provided for training was far less than needed in other studies targeting subtle improvements with passive devices. For example, in both Simpson et al. and Collins et al., participants trained for four sessions, each including more than an hour of locomotion.

Response: Thanks for your comment. Training for long enough time does be necessary, and active devices usually need longer training time than passive ones. In the study of passive ankle exoskeleton by Collins et al., the training time for each condition is 7 min, but in their active device, the training time does be longer. Given that the passive device designed in our study has a lower mass and smaller assistance magnitude, we selected training time of 6 min.

Response to Reviewer #2

1. **Comment:** This paper presents a biarticular passive assistive device based on the concept of exotendon (introduced by van den Bogert). This passive device contributes to hip extension and knee flexion and supports human walking in the late swing and early stance phases. The results show a decrease in Semitendosus muscle without significant changes in other muscles as well as gait kinematic and kinetic behavior.

Response: Thanks for your careful reading.

2. **Comment:** The paper is clear and presents a method for optimizing the device parameters (stiffness and rest length of the spring). However, the main contribution of the paper is not new. The same concept in a comparable design was introduced in a biarticular thigh exosuit in the following article [a]. There are several studies on biarticular muscles and the applications in robotics and assistive technologies including two review papers [b,c] and several other simulation and experimental studies which are ignored by the authors. The only difference between the design in the introduced exosuit in [a] and the device of this paper is that in [a] both two antagonistic biarticular muscles are used to mimic RF and HAM muscles, and this paper uses just HAM-like muscle. If we accept that this is a sufficient contribution for a new

article, the authors at least need to refer to this former study and explain the differences.

Response: Thanks for your recommendations. The recommended articles did broaden our horizons, and several ones have been added as references in our revised manuscript.

Admittedly, the target muscle group in our study is similar to that in the study of Barazesh and Sharbafi (Barazesh and Sharbafi, 2020). But the design thinking is different. For an assistive device, appropriate control is very important. Control strategies can be divided into the model-based and model-free approaches, and both approaches have advantages and disadvantages (Firouzi et al., 2020). The idea of Barazesh and Sharbafi originated in the requirement of balance, as one of three subfunctions during human walking. They conceived a model-based control strategy, where they achieved force modulated compliant hip control framework by length feedback. Then they examined muscle activation, muscle force and the energy cost with the assistance strategy via neuromuscular simulations. Finally, they developed a passive exosuit and conducted an experiment, which demonstrated the success of the exosuit in increasing walking economy and the effectiveness of the simulation results. However, in our study the idea was inspired by joint moment characteristics of walking and passive designs from previous studies. We adopted a model-free control strategy and achieved the assistance based on a predefined gait pattern. To this end, we conducted a preliminary experiment to understand the characteristics of joint moment and assistance produced by the exotendon. Finally, we conducted an experiment and systematically examined the effects of the exotendon on muscle activation, kinematics and kinetics. In total, we thought Barazesh and Sharbafi's study put more focus on the model and simulation while we put more focus on the experiment. We have introduced this publication in *Introduction* part of the revised manuscript.

We have removed the incorrect expression about the biarticular devices. Biarticular muscles and applications of biarticular elements to robotic devices have also been introduced in detail (Page 3 Paragraph 2).

- Comment:** In addition to this significant drawback, the design concept and the details are not solid. The authors neglected the importance of the ratio between hip to knee lever arm which is an important design concept. There are some incomplete argumentations and although I see the importance of the analyses to show the effects of the designed exotendon on supporting human gait, minor improvement in one muscle activation while neglecting some other important

muscles which might be affected such as hip monoarticular muscles (e.g., gluteus maximus) is not justified. Further, the kinetic and kinematic analyses in separate experiments, unnecessary descriptions and unclear figures make this paper inappropriate to be accepted for publication in its current format. See more details in the following.

Response: Thanks for your comments. Lever arm ratio does be an important design parameter, as you pointed out. In our study, the lever arm ratio determines the relative contribution of the assistance in the hip and knee joints. Because the assistive force is equal for two joints, the assistive torque ratio is equal to the lever arm ratio. Theoretically, in order to obtain similar patterns between the assistive torque and the human joint moment (calculated by inverse dynamics) at the hip and knee joints, the lever arm ratio should be set as the ratio of the hip joint moment to knee joint moment (this is to ensure the similarity between assistance patterns and joint moment patterns for all two joints), which is difficult to achieve because the ratio of the hip joint moment to knee joint moment changes with time and the lever arm ratio is difficult to follow the change. So in our study we cannot ensure similar patterns between the assistive torque and human joint moment. Additionally, in our opinion, a configuration with an almost constant lever arm, which was often used in previous studies, may improve the similarity of joint moment pattern, but not necessarily. That is because in this situation the assistive force may not have a similar pattern to the human joint moment (if the lever arm is constant, the assistive torque is dependent on the forces). Taken together, we thought the lever arm ratio is important, but it is difficult to control it as we want. So in our study the lever arm is only used to calculate the assistive torque, and we didn't pay more attention to how to control it. We have added some discussion about the lever arm in the revised manuscript (Page 13 Line 29).

Admittedly, there are some limitations in our study, as introduced in the *Discussion* part (Page 15 Line 27). First, gluteus maximus, as an important hip extensor, was neglected, though theoretically it should show a lower activation. Second, limited by laboratory conditions, the kinematic data and kinetic data were collected separately. Third, the assistance magnitude was not so large, which may contribute to the not large enough improvement. Notably, this improvement was not guaranteed to be the best performance. Next, we are going to overcome these shortcomings and further explore the potential of the exotendon in future studies.

Unnecessary descriptions and figures (about the interface force and assistive torque

calculation method) have been removed according to your advice.

Introduction:

- Comment:** The authors nicely explained the literature on related passive assistive devices and motivated the attention to hip and knee joints. However, they ignored many relevant studies on biarticular muscle both biomechanics and assistive technologies. There are several articles on the design and control of active and passive exosuits using biarticular thigh muscle design (see some references below).

Response: Thanks for pointing this out. We have read the recommended articles and added more introduction to benefits and application of the biarticular configuration to robot and assistive devices (Page 3 Paragraph 2).

- Comment:** The main contribution in this paper is the biarticular design which has several advantages. The authors did not sufficiently motivate this important design feature. It seems that the only reason for designing the device is that the others did not do it which is not correct, as explained above. I encourage the authors to have a better literature review about the role of biarticular muscles in biomechanical and robotic studies to better motivate developing such a device. The following papers could be helpful.

Response: Thanks for your advice. We have reorganized the *Introduction* part.

Methods:

- Comment:** The authors focused on the stiffness and rest length of the artificial muscles (spring), while the hip to knee lever arm ratio plays an important role in specifying the contribution at each joint. In the proposed design, there is no methodology to define the lever arms. Selecting the bottom attachment point around the ankle results in a large variation in the knee lever arm which is not suitable for replicating human joint torques.

Response: Thanks for your insightful comments. The lever arm does be an important design parameter, and the lever arm ratio of the hip to knee will affect the relative contribution of our device to each joint. It is worth noting that the lever arm is just one factor affecting the assistive

torque patterns. As the assistive torque is calculated as the product of the assistive force and the lever arm, patterns of forces and lever arm together determine the assistance patterns. However, these two parameters are of poor controllability for passive devices, and it is difficult to adjust them as desired. The force is affected by the lower limb movement and the spring stiffness. The lever arm will change with the lower limb movement or keep constant via a special configuration. Constant lever arm may simplify the realization of similar patterns because in this situation similar patterns to the human joint moment can be obtained as long as the force patterns are appropriate. But obtaining appropriate force patterns is still difficult. Although using variable stiffness spring can alter the force pattern, altering the stiffness as desired is still difficult. Therefore, the lever arm is just a parameter that we use to calculate assistive torque, and we did not put much focus on trying to adjust it as desired. In addition, that the bottom attachment point locates around the ankle aims to avoid too small lever arm when the leg is fully extended during the late swing phase, which is also one reason that the spring is arranged distally in a previous study (Simpson et al., 2019).

7. **Comment:** The authors assume that the muscle efforts will be decreased which should be verified at the end. However, this is not supported as the hip monoarticular muscles' activations are not measured. Maybe there is an increase in hip flexor muscles (not measured) as the spring acts against it.

Response: Thanks for your comments, and we agree well with your opinion. Gluteus maximus is an important hip extensor. Theoretically, the exotendon will provide assistance in hip extensors. But we put our focus only on the biarticular muscle group and ignored the monoarticular muscles, which is admittedly one limitation of this study. In order to estimate the potential effect on the antagonistic muscle of the assisted muscles, the main muscles of the lower limb were measured, and the rectus femoris, as a hip flexor, was also included.

8. **Comment:** In the second session of the preliminary experiments on the treadmill, the subjects wear the exotendon and the stiffness considered in these experiments will affect the movement and then the force direction. It is not described how the stiffness and rest length in these experiments are selected. For example, if the springs are too soft or too stiff, the results will be different. Further, doing separate experiments on the treadmill and walkway for measuring the kinematics and kinetics is not justified. We know that treadmill walking and ground walking

are not similar. In addition, inscribing step frequency with a metronome does not mean similar speeds the step length might be different.

Response: Thanks for your comments. The second session of the preliminary experiments on the treadmill aimed to obtain the lever arm change over a gait cycle and the distance between the two attachment points of the exotendon in a gait cycle (this distance will affect the assistive force because we used a constant stiffness spring). The lever arm is calculated via the force direction (determined by two markers attached to the two ends of the spring) and the joint centre (determined by markers attached to the great trochanter and the lateral condyle of the femur, respectively).

The rest length affects the accuracy of the lever arm measurement. That's because the rest length affects the time range of the assistive force. When the exotendon is slack, the force direction measured by two markers on the two ends of the spring is incorrect, and thus the lever arm measurement is incorrect. Therefore, we need to ensure that the time range of producing force is appropriate (i.e., hip extension moment lasts till the middle stance phase, so the assistive force should not disappear until the middle stance phase). To this end, we let participants keep a pose similar to that in the middle stance phase (stand with the knee slightly flexed) and the exotendon was slightly preloaded.

The distance between two attachment points is calculated via two markers attached to the body. If we suppose that the human movement is not affected, the distance between these two attachment points will be not affected by the stiffness and the rest length. The spring stiffness affects the magnitude of the force, and large forces may affect human movement. In order to minimize the effect on human movement, we selected a stiffness of about 600N/m. We have added these descriptions. (Page 6 Lines 2, 15)

Admittedly, doing separate experiments on the treadmill and walkway for measuring the kinematics and kinetics is not an ideal way, and we had better collect these data at the same time. But limited by the laboratory conditions we cannot achieve it. So we adopted a method that was usually adopted before the treadmill with the built-in force plate appears. Controlling the step frequency with a metronome is a way to control the walking speed in order to make the speed close to that in the treadmill walking as far as possible. But indeed it cannot ensure it. This measurement may affect the optimization results, but we thought that the effect is limited

and acceptable.

9. **Comment:** The description of the joint torque calculation is too long with unnecessary formulations in appendix 1. All equations are trivial and not necessary. For example the weird term "and the assistive torque produced by one Newton force that was calculated from the preliminary experiment." can be replaced by "and the lever arm".

Response: Thanks for pointing this out. We have removed the appendix and replaced the redundant expression according to your suggestion (Page 6 Line 25; Page 7 Line 12).

10. **Comment:** Equation 2.4 is used for optimization, but in the results, it is not used for evaluating the quality of the device in decreasing the muscle contribution. It seems that Fig. 5.b and Fig. 7 show the total moment, which is the summation of the human muscle and the springs as calculated by the GRF through inverse dynamics.

Response: Thanks for your comment. In the experiment results, we did not show the reduced biological joint moment directly. Limited by the laboratory conditions, the assistive torque was measured during the treadmill walking while the joint moment calculated by inverse dynamics was measured during the force plate walking. We thought it may be a little inappropriate to get the biological joint moment by subtracting the assistive torque from the joint moment calculated by inverse dynamics (i.e., human biological joint moment + assistive torque). Therefore, we evaluated the performance of the device indirectly by demonstrating that the assistive device did not affect the joint moment calculated from inverse dynamics (i.e., human biological joint moment + assistive torque). Then provided with the assistive torque, the biological joint moment will decrease.

11. **Comment:** It seems that the maximum stiffness in the range is found as the optimal value. Why? can you explain what happens? What if the range is increased. The equation can be solved analytically to find the global minimum and the optimization result can be found with a closed-form equation—no need for GA optimization to find the local minimum in the range, although it is acceptable.

Response: Thanks for your question. As we pointed in *Methods* part, in order to limit the effect of too large force on the kinematics and kinetics, we limited the spring stiffness range to 0-3kN/m. If the range was not limited, it is easy to imagine that the optimized assistive torque

will be comparable to the human joint moment (calculated by inverse dynamics) during the assistance phases. That is too large, because for most previous studies, the assistance magnitude accounts for approximately 10-50% of the human joint moment (for example, passive ankle exoskeleton designed by Collins et al. provides assistance of a magnitude of about 10-25% of the ankle joint moment), and the optimal assistance magnitude of some studies is also in this range. Therefore, when the stiffness range is small, the optimized stiffness will always be the upper limit. We have explained the reason in the revised manuscript (Page 8 Line 2).

We tried to use the analytical method to solve the optimization problem. But we have not figured it out because we find it difficult to calculate integrals with absolute values.

12. **Comment:** The following sentence is not precise as it will move forward again at the next push-off:

"The gait cycle was segmented by the marker on the right heel, where heel-strike was thought to occur when the marker moved to the anterior most position."

Response: We are sorry for our unclear expression, and there appears to be a misunderstanding. During the treadmill walking, the foot moves to the forward most position when heel-strike occurs. According to the phenomenon we segmented the gait cycle.

13. **Comment:** To calculate the total joint moment, why don't you use the integral of the absolute of the joint moment (similar to 2.4) as the negative and positive torques might compensate each other without using absolute.

Response: We are sorry for our incorrect expression. The total joint moment was not calculated as the integral of the joint moment directly, but the integral of the absolute value of joint moment, as you stated. We have revised the expression (Page 10 Line 9).

Results:

14. **Comment:** The figures at the end of the paper are all corrupted. All the texts are mixed with figures and the line numbers, and legends.

Response: We are sorry for the inconvenience. It seems that it will appear when we upload the figures separately to the submission system, because the system will add the texts and the line numbers automatically.

15. **Comment:** The authors mentioned the discrepancy between the timing of the Hip and Knee peak torques prior to heel-strike (Fig.2). This is different from the human torque patterns (Fig. 5b) which might result from the lever arm adjustment.

Response: We agree well with your opinion. As mentioned above, the lever arm adjustment over a gait cycle is just one reason for the inconsistency in assistive torque patterns and the human torque patterns. And the other reason is the force change in a gait cycle.

16. **Comment:** Which one is correct? Scaling factor or scalar factor?

Response: Thanks for your question. It should be scaling factor. Here we want to compare the similarity of two curves, which includes the pattern similarity and the amplitude similarity. The scaling factor reflects the amplitude similarity. We have revised the incorrect expression (Page 11 Line 31).

17. **Comment:** It is stated that "the maximum extension of the hip increased by 6.9 degrees", but this is not visible in the figure. The difference between EXO and CON experiments in hip flexion (negative peak) is more pronounced (in both angle and moment figures) than the slightly reduced extension peak explained by the authors. What is the reason for this increase flexion torque in the region that the exo does not contribute?

Response: Thanks for your comment. We have discussed these results in more detail in the revised manuscript (Page 14 Line 26; Page 15 Line 15). Maybe there is a misunderstanding. We guess what you want us to explain is the difference appearing in the push-off (the unassisted phase) because in this phase the difference is more pronounced. If so, the hip joint is not in the maximum flexion, but in maximum extension.

During the time the exotendon works, the hip flexion decreases. It is not unexpected because the exotendon provides hip extension torque. But during push-off (the unassisted phase), the hip extension increases significantly. We thought that this can be considered as a compensation mechanism that human body makes an adjustment to change the joint range of motion (ROM) as little as possible. Smaller change of the joint ROM will contribute to the minimally affected step parameters. That's to say, participants tended to keep the step parameters invariant.

Since the joint moments were calculated by the inverse dynamics, the change of inertial force caused by the altered kinematics may contribute to the slight difference. In addition,

because the difference in the joint moment mainly occurs in the stance phase, the change of ground reaction force may also be an important factor. But why the human body made such adjustment remains unclear. We expect to design experiments and further investigate this phenomenon in the future study.

18. **Comment:** Figs 8-10 can be removed. Not informative! Fig. 10 is not even described. What is the orange line. It is not readable and it is not clear why the authors show it.

Response: Thanks for pointing this out. We have removed these figures according to your advice.

Discussion:

19. **Comment:** The goal of the design was to minimize the total biological joint moment of the hip and knee, as mentioned also at the beginning of the discussion. However, it was not supported by the results and is not discussed in the discussions.

Response: Thanks for pointing this out. As mentioned above, here we showed the reduced biological joint moment indirectly. Limited by the laboratory conditions, the assistive torque was measured from the treadmill walking trials, and the human joint moment (in the EXO condition, human joint moment = human biological joint moment + assistive torque) was measured via inverse dynamics from the force plate walking trials. The speed and the step frequency were different between the treadmill walking and the force plate walking. So we think it a little inappropriate to get the biological joint moment by subtracting the assistive torque from the human joint moment. Here we just want to convey that the exotendon will not affect the human joint moment, which indirectly shows that the biological joint moment will decrease when the joint is provided with external assistive torque. We have added this in the revised manuscript (Page 15 Line 22).

20. **Comment:** Decreasing the interface torque over time is expected due to the method of attaching the two sides of the exotendon to the body. This is a known phenomenon and it is not worth to be part of the results and also be discussed in detail in Sec. 4. There are also methods to prevent it, such as using a strap below the shoe sole to prevent the lower segment movement.

Response: Thanks for pointing this out, and we have removed this part.

21. **Comment:** When the authors discussed other options, they argue "Actually, larger magnitude does not always produce better performance [7, 18, 35]." This is correct, but it does not mean that we should not examine higher magnitudes because it might provide inconvenience. Further, the lever arm effects should be analysed more carefully and not just the magnitude.

Response: Thanks for your comment, and we agree with you. Determining the optimum assistance via experiment is still the most persuasive way, so examining a higher magnitude does be necessary and significant. We know that the actual optimum assistance in this study may be larger. But finding the optimum assistance was not the focus of this study. This study is more of a proof-of-concept study. So we selected small but effective assistance to ensure the comfort of the wearers.

The lever arm and the variable stiffness spring can affect the assistive torque patterns. We have made necessary discussion (Page 13 Line 29).

22. **Comment:** When the authors discuss the effects on other hip extensors and knee flexor muscles, they explain gastrocnemius, but not Gluteus Maximus, which was not even measured. This is one of the important muscles which should be analyzed.

Response: Thanks for pointing this out. Admittedly, this was a miss in the experiment design. Gluteus maximus, as an important hip extensor, theoretically should have a lower activation. But we did not measure it, which was one limitation of this study.

23. **Comment:** The following statement is also not completely accurate as in the first 10% of the stance phase, the spring supports knee torque:

"i.e., the exotendon will hinder the knee joint movement during the late period of the early stance phase."

Response: We are sorry for our unclear expression. The assistance interval is slightly larger than 10% of the gait cycle, so the exotendon will hinder the knee joint movement during the late period of the assistance (i.e., approaching the middle stance phase). We have revised the expression (Page 14 Line 9).

24. **Comment:** In the following sentence the authors discuss the knee flexion. What about the hip flexion (peak) which shows more pronounced difference.

"Similarly, the knee flexion decreased during the unassisted phase, which may be related to the necessity of altering the ground clearance due to change in the step length."

Response: Thanks for your comment. As mentioned above, we guess there may be a misunderstanding. We have made necessary discussion about the difference in more detail (Page 14 Line 26).

25. **Comment:** The following sentence is not correct as gait variation and stability are two different concepts. An assistive device might change the gait and make it even more stable.

"In addition, in this study the gait variability did not show a significant difference between conditions, which suggested that the exotendon did not introduce the stability problems."

Response: There appears to be a misunderstanding. Gait variability shows the step-to-step fluctuations, and it was calculated as the standard deviation of the gait parameter in this study.

26. **Comment:** Later the authors say, ". In our study the lightweight design and small assistance may account for the unaffected stability." This is also not strong as the same device with too stiff spring might result in a different outcome.

Response: We agree with your opinion. Actually, your opinion is also what we want to convey. The unaffected stability may be attributed to the lightweight design small assistance. Stiffer spring will cause larger assistance, and the result may not remain unaffected.

Once again, we thank you for the time you put in reviewing our paper and look forward to meeting your expectations.

Barazesh, H. & Sharbafi, M. A. 2020. A biarticular passive exosuit to support balance control can reduce metabolic cost of walking. *Bioinspiration & Biomimetics*, 15, 036009.

Firouzi, V., Davoodi, A., Bahrami, F. & Sharbafi, M. A. 2020. From a biological template model to gait assistance with an exosuit. *BioRxiv*.

Simpson, C. S., Welker, C. G., Uhlrich, S. D., Sketch, S. M., Jackson, R. W., Delp, S. L., Collins, S. H., Selinger, J. C. & Hawkes, E. W. 2019. Connecting the legs with a spring improves human running economy. *Journal of Experimental Biology*, 222, jeb202895.

Appendix B

Dear editors and reviewers,

Thanks for giving us the opportunity to resubmit our revised manuscript entitled “A portable exotendon assisting hip and knee joints reduces muscular burden during walking” to the journal “Royal Society Open Science” for your reconsideration of its suitability for publication. We appreciate the time and effort that you dedicated to providing feedback on our manuscript. To solve problems proposed by reviewers, we have redesigned the experiment. The main change in the experiment design includes four aspects. First, all trials were conducted on an instrumented treadmill embedded with force plates. In our previous experiment, the trials were divided into treadmill walking trials and force plate walking trials. This can solve problems caused by the asynchronous measurement of kinematics and kinetics or difference in treadmill walking and overground walking. Additionally, the gait segmentation used the same method for all data, i.e., ground reaction force based method. Second, we selected three different assistance levels. Limited by the assumption of the optimization, the stiffness cannot be obtained by the optimization as desired. So we did not optimize the stiffness anymore. Third, we added the measurement of gluteus maximus which was ignored in our previous experiment design. Finally, the optimization that determines the physical parameters of exotendon was performed for every subject. In our previous study, we selected a subset of the subjects to obtain the data required for the optimization, which ignored the individual differences.

Now, we have rewritten most contents of the manuscript according to reviewers’ comments. Besides the revision due to the altered experiment design, we also carefully summarize the state-of-the-art studies in the Introduction part and have a further discussion with the experimental results in the Discussion part. All changes in the revised manuscript are marked in blue. Please see below, in blue, for a point-by-point response to reviewers’ comments and concerns.

Yours sincerely,

Caihua Xiong

Institute of Rehabilitation and Medical Robotics, Huazhong University of Science and Technology,
Wuhan, Hubei 430074, China

E-mail: chxiong@hust.edu.cn

Response to Reviewer #2

1. **Comment:** Major concern regarding the contributions:

In the previous review I mentioned that there are several relevant studies that should be read and considered to clarify the contribution of this work to the state-of-the-art. Although few new articles are cited in the revised version, it is done in a very superficial way. The key concept of the design of the exo which is using a biarticular spring is not motivated and is not linked to other studies. The authors described the contributions of this study compared to [22] which use the same design concept in the response letter, but in the paper adding two neutral sentences seems to be an attempt to hide this close relation between these two studies. I believe if the authors honestly describe the differences to the state-of-the-art, it elates the quality of the paper and not lowers your work. PLEASE carefully place your work in the stat-of-the-art.

Response: Thanks for your comments. We have rewritten the *Introduction* part to review the state-of-the-art studies more comprehensively. The benefits of biarticular muscles in human locomotion are introduced and the inspired biarticular designs are reviewed (Page 3, Paragraph 2). In the previous responses, we have detailedly explained the difference between our study with the other study which introduced an exosuit to support the hamstring muscles and the rectus femoris. All in all, we thought their study put more focus on the model and simulation while we put more focus on the experiment and the biomechanical effects. We have introduced their contributions in more detail (Page 3, Line 37) and summarized the differences between ours and theirs (Page 4, Line 21).

2. **Comment:** The motivation of the work and the hypothesis is still not clear. Why do you use biarticular springs? After writing several paragraphs about passive and active exos, the last paragraph of into says "In this study, we aimed at further investigating the biarticular assistance strategy and examining the effects of the strategy on human movements". Why biarticular? what is the hypothesis behind it?

Response: Thanks for your comments. We have rewritten the *Introduction* part. We learned from previous studies that successful assistive device design can be achieved from several aspects. There are three reasons for the biarticular design in our study. Firstly, the

biarticular muscle-tendon units play important roles in improving human locomotion efficiency. For example, they support the energy storage and return similar to the monoarticular ones. They also promote the energy transfer from the proximal joint to the distal joint. These mechanisms inspired the biarticular designs. Secondly, recent studies have shown the potential of the biarticular assistance strategies. Biarticular knee-ankle assistance has been found higher metabolic reduction than monoarticular ankle assistance (Malcolm et al., 2018). Many other studies also succeed in improving the economy during various kinds of locomotion using biarticular designs. These encourage us to further explore the biarticular devices. Thirdly, in this study, the assisted swing leg retraction needs hip extension assistance and knee flexion assistance. This requirement is the theoretical basis that we can have a biarticular design, and the design can be simply achieved by a design mimicking function of the hamstring muscles. (Page 3, Paragraph 2; Page 4, Paragraph 4)

3. **Comment:** The authors referred to the work from van dijk 2014 [20] which was another implementation of the exotendon. You correctly mentioned that "The biological joint moment significantly decreases.", but you did not mention that the conclusion of [20] in the invalidity of the exotendon concept as the metabolic cost is not reduced. It seems that one of the main findings in your paper is also the reduction of biological torque which is also not supported by your outcomes (see below for detail). If we accept that you can reduce biological joint moments, it is still not sufficient to support the effectiveness of the exo.

Response: Thanks for pointing this out. We wanted to emphasize the role of the exoskeleton in reducing biological joint moment in the original manuscript, which was indeed incomplete. We have introduced the study more accurately (Page 4, Line 10).

The goal of our study is not to reduce the biological joint moment, but to reduce muscle effort. Reducing the biological joint moment is just a way to achieve the goal. Many previous studies succeeded by exerting the assistive torque to replace part of the human joint moment. Therefore, we took the biological joint moment as the optimization objective in the optimization process. Due to complexity of biological system, we further demonstrated experimentally that the biological joint moment did decrease as expected. In the revised manuscript we directly evaluated the difference in the biological joint moment among conditions, which can support

our hypothesis.

4. **Comment:** The lever arm is one parameter but a crucial one that can change the ratio of the generated torque at hip and knee and it is not reasonable to not discuss it because it is one parameter. The response of the authors is not convincing at all.

What is presented in the discussion does not address the comment. The main point is not about having constant or variable lever arms. The authors should try to approach human-like lever arm ratio to generate consistent joint torques to humans. Even if it is variable, simple solutions like finding appropriate attachment points, mechanisms like cam or guides can help provide similar hip to knee lever arm RATIO which is more important than the value of the lever arm. If the ratio is too different from what observed in the human body, the contribution of the exo will contradict human muscle actuation. The authors need to discuss this effect instead of saying we use a variable lever arm because we cannot keep the lever arm fixed!

Response: Thanks for your comments. We agree that the lever arm ratio will determine the ratio of the generated torque at the hip and knee. But we don't figure out why approaching a human-like lever arm ratio can ensure consistency with human joint moments. There are two puzzling problems. First, according to our understanding, different lever arm ratios will cause different relative assistance magnitudes. (Here the relative assistance magnitude means the ratio of assistive torque to the human joint moment) For example, suppose that the ratio of the hip to knee joint moment is 1.5, if the lever arm ratio is set to 1.5, then the relative assistance magnitudes of the hip and knee are the same. If the lever arm ratio is not 1.5, the relative assistance magnitude will be different between hip and knee. Therefore, it seems that the lever arm ratio just causes different relative assistance magnitudes of hip and knee. We think that only when the assistance magnitude is too large, inappropriate lever arm ratio will hinder joint movement, as you stated. In this case, inappropriate lever arm ratio is easy to cause relative assistive magnitude of more than 1, thus hindering joint movement. The assistance magnitude in our study is relatively small, so we think a not human-like lever arm ratio is acceptable. Second, we guess the human-like ratio you mentioned is the muscle-like lever arm ratio. If so, we may think it a little inappropriate. The human joint moment calculated by the inverse dynamics is not the generated torque of a certain muscle, but the superposition of all muscles

torque. Therefore, we think whether simply adopting a certain lever arm ratio like certain muscles can benefit the performances of devices remains unclear.

We selected the distal shank as the lower attachment point in our design, which was mainly based on two considerations. First, it can increase the lever arm relative to the hip and knee joints. Because our device contains no rigid structures, the system stiffness of the device may be limited. In order to obtain large enough assistance magnitude, the lever arm should be as large as possible. Second, anchor the device to the locations on the body of high stiffness can benefit the system stiffness. It can also minimize the movement of the ankle strap when applying the assistive force. Certainly, this arrangement will increase the variation in lever arm, which is disadvantageous. But from the perspective of the assistive torque in our experiment results (figure 2), the assistive force has a larger effect on the assistive torque patterns, and the effects of variation of the lever arm is limited.

It should be emphasized that, in our opinion, it is not clear whether approaching the human-like lever arm ratio can produce better performances when the assistance magnitude is not so large, and kinds of attempts are worthy of recognition. For example, an exoskeleton removing joint negative power (Shepertycky et al., 2021) and an exotendon connecting two legs (Simpson, Welker et al. 2019) are attached to the body distally. Certainly, study approaching the human-like lever arm ratio also exists (Barazesh and Sharbafi, 2020). All these studies are successful.

We have summarized the effect of the lever arm ratio on the results in our study (Page 12, Line 6).

5. **Comment:** The authors just responded to the previous comments superficially by saying this is the limitation of our study and did not modify the paper accordingly they even did not describe them carefully in the discussion. The authors should know that the goal of criticizing the paper is not to convince the reviewer, but to improve the quality of the paper which is not changed (except adding two references and few neutral sentences in the intro) from the previous version.
Response: Thanks for your comments. We have added the measurement of the gluteus maximus in our new experiment. We have revised most of the contents of the manuscript in the current version.

6. **Comment:** Instead of addressing the comments the authors just repeated what was written in the paper in the response letter! The way of calculating the lever arm was already described in the paper. The problem is that it was not considered in the design and after doing the experiments it was not considered in explaining the findings.

Response: Thanks for your comments, and maybe there was a misunderstanding. The original comments were about the effects of selection of exotendon physical parameters (i.e., resting length and stiffness) in the preliminary experiment on the results. Because the preliminary experiment was performed to collect data required for the optimization process (i.e., joint moment, exotendon length change in a gait cycle and moment arm change in a gait cycle), the effects on the results were mainly caused by affecting the measurements of these data. Therefore, we analysed the effects on these data measurements. Selection of exotendon physical parameters in the preliminary experiment affects the moment arm measurement mostly because whether the exotendon is in tension will affect the correctness of the moment arm measurement. Then based on the calculation process of the moment arm, we introduced how the effects are caused in detail. Based on these considerations, finally, we explained how we determined the exotendon physical parameters in the preliminary experiment (Page 6, Line 22).

7. Again the response letter says "... But indeed it cannot ensure it. This measurement may affect the optimization results, but we thought that the effect is limited and acceptable." This must be discussed in the paper, not in the response letter

Response: In the previous responses we discussed the effect of asynchronous measurements of kinematics and kinetics on the results. We have re-experimented using an instrumented treadmill. Problems resulting from the asynchronous measurements of kinematics and kinetics have been solved in our new experiment.

8. **Comment:** The argumentation about decreasing biological joint torque is not correct as in some conditions the Exo torque might oppose the biological torque. This means even with similar total joint torque the biological torque could increase with exo. If we accept the argumentation from the authors, even if we use a very stiff spring or damper and ask the subjects to keep the same kinematic behavior, and the GRF stays the same, the biological torque will reduce which

we know it is not correct. As it is one of the key outcomes of the paper it needs to be scientifically supported.

Response: Thanks for your comments. We agree with your opinion. In the original manuscript we did not measure the assistive torque and human joint moment simultaneously, so we did not directly calculate the biological joint moment. In this revised manuscript we have re-experimented using an instrumented treadmill and collected the data simultaneously. And we estimated the change in biological joint moment directly (Page 9, Paragraph 2).

9. **Comment:** I understand the argumentation about partial support of the joint torques. However, it is not emergent true to say if the range of stiffness is not limited, the optimized assistive torque will be comparable to human joint torque because you are trying to approach both hip and knee torque just by one biarticular passive spring. Since the knee and hip torques are reciprocal just in part of the gait cycle, optimization of HAM-like stiffness should not necessarily give similar joint moments. If we accept this reasoning why do you need optimization?

Response: Thanks for pointing this out. We agree with your opinion that using just by one spring cannot ensure that the assistive torque is comparable to human joint moment for both joints. Because we use the biological joint moment as the optimization target, if the stiffness is not limited, the assistive torque will be very large. Therefore, we limited the range of stiffness to a relatively small value. But this will result in the other question that the optimized stiffness is always the upper limit, and thus the optimization process cannot select a relatively better stiffness. For this reason, in our revised experiment design we did not optimize the spring stiffness anymore but just select three kinds of stiffness, and we optimized the resting length for each stiffness. (Page 6, Paragraph 2)

10. **Comment:** In this sentence, it is not clear that this method is used for treadmill walking. Moreover, the leg moves backward prior to heelstrike which is called swing leg retraction. This means that the most anterior position of the heel marker is not exactly the heel strike.

Response: Thanks for your comment. We once compared the marker-based method with the force-based method in the force plate walking. There will be deviations of about 1-3 frames (sampling rate: 100 Hz). But we thought it just offsets the curves during a gait cycle and will

not affect the results in the original manuscript. Certainly, in the current revised manuscript, we have re-experimented using an instrumented treadmill, and the heel-strike was segmented based on the ground reaction force (Page 7, Line 12).

11. **Comment:** As I explained above the argumentation about reduced biological torque is not correct. Even if you show it, still the exo can be unhelpful as shown in the study from van dijk. Then, what we can learn from this study?

Response: Thanks for your question. As mentioned above, reducing the biological joint moment is just a way to achieve the goal of reducing muscle efforts, which is similar to many recent studies on active devices. These active devices also mainly provide the external assistive torque to replace the human joint moment and thus reduce the biological joint moment. We estimated the effectiveness of our exotendon mainly by the activation of the assisted muscles. We showed the decreased biological joint moment to verify the design principles. Certainly, as you mentioned, only the reduced biological joint moment cannot demonstrate the effectiveness of the assistance. So we also measured the muscle activation which is also an estimation of the assistive device performances. Meanwhile, we also examined the effect of the assistance on the kinematics.

12. **Comment:** It is clear what gait variability means. The stability term is not clear in your argumentation. Stability has a clear meaning in control engineering and gait analyses. If you mean gait stability (repetitive movement without falling like a stable limit cycle) it cannot be identified by the variability between different experimental scenarios but within one specific experiment. If you mean the similarity to the normal gait you can argue like this, but mainly the first meaning is realized from stability.

Response: We are sorry for our unclear expression. The former meaning is what we want to convey. Gait variability shows the step-to-step fluctuations, and it is calculated as the standard deviation of the gait parameter in this study. It is calculated for each walking trial. For example, for a certain subject, the step length variability in the CON condition is calculated as the standard deviation of step length data from the selected time range in the CON condition. (Page 7, Line 38)

Barazesh, H. & Sharbafi, M. A. 2020. A biarticular passive exosuit to support balance control can reduce metabolic cost of walking. *Bioinspiration & Biomimetics*, 15, 036009.

Malcolm, P., Galle, S., Derave, W. & De Clercq, D. 2018. Bi-articular knee-ankle-foot exoskeleton produces higher metabolic cost reduction than weight-matched mono-articular exoskeleton. *Frontiers in neuroscience*, 12, 69.

Shepertycky, M., Burton, S., Dickson, A., Liu, Y.-F. & Li, Q. 2021. Removing energy with an exoskeleton reduces the metabolic cost of walking. *Science*, 372, 957-960.

Simpson, C. S., C. G. Welker, S. D. Uhlrich, S. M. Sketch, R. W. Jackson, S. L. Delp, S. H. Collins, J. C. Selinger and E. W. Hawkes 2019 Connecting the legs with a spring improves human running economy. *Journal of Experimental Biology* **222**(17): jeb202895.

Appendix C

Dear editors and reviewers,

We are submitting our revised manuscript entitled “A portable exotendon assisting hip and knee joints reduces muscular burden during walking” to the journal “Royal Society Open Science” for your reconsideration of its suitability for publication. We appreciate the time and effort that you dedicated to providing feedback on our manuscript. We have incorporated the suggestions made by the reviewers, and the whole manuscript has been edited by Charlesworth Author Services, a language editing service provider (see appendix for the certificate of editing). All changes in the revised manuscript are marked in blue. Please see below, in blue, for a point-by-point response to reviewers’ comments and concerns.

Yours sincerely,

Caihua Xiong

Institute of Rehabilitation and Medical Robotics, Huazhong University of Science and Technology,

Wuhan, Hubei 430074, China

e-mail: chxiong@hust.edu.cn

Response to Reviewer #2

1. **Comment:** After the revision, the consistency of the paper needs to be checked. It seems that the authors rush in revising some parts which could be improved. For example, this sentence seems isolated and not well presented. "Secondly, assistance parameters are well selected."

Response: Thanks for pointing this out. We have revised the manuscript to address your concerns and hope that it is now clearer. The sentence that you pointed out has been replaced with "Researchers try to maximize the benefits from external assistance mainly in two ways, i.e., applying assistance to subtasks of effort, such as redirection of the centre of mass, and selecting appropriate assistance parameters. ... Additionally, appropriate assistance parameters also play an important role in maximizing the benefits from the external assistance." (Page 3 Paragraph 1).

2. **Comment:** I could say that the writing quality is lowered in some cases. Again, revising the paper by a native speaker is suggested. Here are some examples (not limited to that can be improved in the introduction section, but they are not limited to these points.
 - a) A biarticular knee-ankle exoskeleton has been found a higher reduction in metabolic cost than the monoarticular assistance.
 - b) More recently, exploring tasks of not too much effort is also popular, such as swing leg deceleration.

Response: Thanks for your suggestion. This manuscript has been carefully revised by a language editing service provider (Charlesworth Author Services, see appendix for the certificate of editing). We have carefully scrutinized the manuscript, and made corresponding revisions including some typos, grammatical errors and long sentences, etc. The two sentences pointed out by you have also been revised as follow,

"A biarticular knee-ankle exoskeleton achieves a higher reduction in the metabolic cost than the monoarticular assistance." (Page 3 Paragraph 2)

"More recently, it is also popular to assist in subtasks of not too much effort, such as swing leg deceleration." (Page 3 Paragraph 4)

3. **Comment:** The rest length adjustment can be described and discussed more.

Response: Thanks for your advice. We have added the following sentences in the main text:

“Physical parameters of the device include the stiffness and the resting length. The stiffness is adjusted by selecting springs of different stiffness, and the resting length is adjusted by the Velcro.”

“According to the above calculation, the resting length affects the timings and magnitudes of the assistive forces, and short resting length will cause large time ranges and magnitudes. The spring stiffness only affects the assistance magnitudes.”

We would like to thank the editors and reviewers again for taking the time to review our manuscript.

Appendix

Figure. Certificate of editing

EDITORIAL CERTIFICATE

This document certifies that the manuscript below was edited for correct English language usage, grammar, punctuation and spelling by qualified native English speaking editors at Charlesworth Author Services.

Paper Title:

A portable extotendon assisting hip and knee joints reduces muscular burden during walking

Author:

Longfei Cheng

Date certificate issued:

October 06, 2021

cwauthors.com